# Atomically Thin 2D van der Waals Magnetic Materials: Fabrications, Structure, Magnetic Properties and Applications

**Wei He [1,2], Lingling Kong [1,2], Weina Zhao [3,\*] and Peng Yu [1,2,\*]**

1 State Key Laboratory of Optoelectronic Materials and Technologies, School of Materials Science and Engineering, Sun Yat-sen University, Guangzhou 510275, China; hewei56@mail2.sysu.edu.cn (W.H.); konglling33@163.com (L.K.)
2 Guangzhou Key Laboratory of Flexible Electronics Materials and Wearable Devices, School of Materials Science and Engineering, Sun Yat-sen University, Guangzhou 510275, China
3 Guangdong Key Laboratory of Environmental Catalysis and Health Risk Control, School of Environmental Science and Engineering, Institute of Environmental Health and Pollution Control, Guangdong University of Technology, Guangzhou 510006, China
\* Correspondences: zhaowngd@gdut.edu.cn (W.Z.); yupeng9@mail.sysu.edu.cn (P.Y.)

**Abstract:** Two-dimensional (2D) van der Waals (vdW) magnetic materials are considered to be ideal candidates for the fabrication of spintronic devices because of their low dimensionality, allowing the quantization of electronic states and more degrees of freedom for device modulation. With the discovery of few-layer $Cr_2Ge_2Te_6$ and monolayer $CrI_3$ ferromagnets, the magnetism of 2D vdW materials is becoming a research focus in the fields of material science and physics. In theory, taking the Heisenberg model with finite-range exchange interactions as an example, low dimensionality and ferromagnetism are in competition. In other words, it is difficult for 2D materials to maintain their magnetism. However, the introduction of anisotropy in 2D magnetic materials enables the realization of long-range ferromagnetic order in atomically layered materials, which may offer new effective means for the design of 2D ferromagnets with high Curie temperature. Herein, current advances in the field of 2D vdW magnetic crystals, as well as intrinsic and induced ferromagnetism or antiferromagnetism, physical properties, device fabrication, and potential applications, are briefly summarized and discussed.

**Keywords:** two-dimensional materials; ferromagnetism; antiferromagnetism; van der Waals magnet

## 1. Introduction

Although research on low-dimensional ferromagnets can be traced back to the early 1970s, due to the limitations of material synthesis technologies, the investigation of ferromagnets was mostly focused on bulk materials for quite a long time [1–3]. In recent years, the development of epitaxial synthesis and exfoliation techniques has greatly advanced the research of two-dimensional (2D) materials. Various kinds of 2D materials, such as transition metal dichalcogenides (TMDs), ternary transition metal compounds ($ABX_3$), transition metal carbide/nitrides (MXene), and their potential related applications, have been reported in succession. In particular, materials that possess 2D ferromagnetism developed rapidly [4], which is of great importance for highly sensitive magnetic measurement methods [5–7]. Ferromagnetism originates from the intrinsic magnetic dipole moment or the parallel arrangement of electron spin in the crystal, which allows for spontaneous magnetization even without an external magnetic field.

For the field of 2D magnetism, there are several key theoretical concepts and frameworks; for instance, the Ising transition theory [8], the Berezinskii–Kosterlitz–Thouess (BKT) transition of the XY model [9,10], and the Mermin–Wagner theorem [11]. According

to the Mermin–Wagner theorem, short-range isotropic interaction cannot realize long-range continuous rotational symmetry in low-dimensional systems, as demonstrated by the 2D isotropic Heisenberg model. This is mainly due to the enhancement of 2D fluctuations, which induces unsustainable order and symmetry breaking. Nevertheless, some earlier theoretical research based on 2D membranes revealed that anharmonic coupling can suppress such fluctuations [12–14]. According to Ising transition theory, it is practicable to generate a low energy gap by introducing anisotropy, which also enhances the stability of long-range correlation in 2D materials to establish long-range order. This indicates that low symmetry of 2D materials is necessary to realize ferromagnetism in a low-dimensionality system. Meanwhile, the existence of internal enhancement and new potential fluctuations, theoretically, may also result in other order types, such as superfluid order, superconducting order, and topologic order [10,11]. However, the introduction of anisotropy requires external effects, for example, an external magnetic field, by introducing defects to reduce the symmetry, etc. [15,16].

Inspiringly, Xu and Zhang et al. discovered ferromagnetic behavior in atomically thin $CrI_3$ [15] and $Cr_2Ge_2Te_6$ [17], opening the door for investigating 2D intrinsic ferromagnets with high Curie temperature ($T_C$). Notably, the layered structure of 2D materials brings many variables to its magnetic properties. In other words, the ferromagnetism of some 2D intrinsic ferromagnets may be layer-dependent, like $PtSe_2$, indicating that it is feasible to modulate ferromagnetic properties, especially the Curie temperature, by adjusting the number of layers or interlayer doping. Significantly, 2D intrinsic ferromagnets provide a great way for researchers to investigate various quantum effects and electron spin, such as the magnetic proximity effect [18–21], anomalous Hall effect [22,23], circular optical polarization [24,25], and so on, all of which are considered to be key to semiconductor self-spin electric devices.

2D magnetic materials are a vital part of the 2D family. In this review, we will focus on the synthesis, structure, ferromagnetism (or anti-ferromagnetism), and related magnetic phenomena of recently reported 2D magnetic materials. We summarize the current research status of 2D magnetic materials and describe prospective challenges, aiming to provide readers with a deeper understanding of this field. In addition, in view of the current challenges in this field, we give some constructive suggestions to provide helpful directions for future research. We believe that comprehensive insight into 2D magnetism is highly conducive, not only to the development of magnetic engineering, but also to expand the application of such materials in spin electronics and related devices.

## 2. Origin of Magnetism

The magnetism of materials, in essence, is a purely quantum mechanical effect, which can be explained by electron spin and the Pauli exclusion principle [26–28]. Electronic spin, as well as its orbital angular momentum, will generate a dipole moment and a magnetic field. In the vast majority of cases, because electrons all exist in pairs with opposite spin, the total dipole magnetic moment of all electrons in a material is zero. Only those materials with partially filled shells will generate a net magnetic moment, from which we can observe a macroscopic magnetic field. According to Hund's rules, the first few electrons in a shell tend to have the same spin, thereby increasing the total dipole moment. According to the Pauli exclusion principle, two electrons with the same spin cannot exist in one orbital. Therefore, as a consequence of the overlapping outer valance electron orbitals from adjacent atoms, the parallel-spin state will have more dispersed electric charge distributions in space than the anti-parallel-spin (opposite-spin) state. In other words, the parallel-spin state has lower electrostatic energy compared to the anti-parallel-spin state; that is, the parallel-spin state is more stable. In quantum mechanics, when two neighboring atoms have unpaired electrons, this effect induces the electronic spin state to tend to a parallel-spin state, which is called an "exchange interaction". Wertheim et al. calculated the range of exchange interaction in iron alloys, which is very short [29]. B. Coqblin and J. R. Schrieffer carried out complex computations of exchange interactions in alloys with

cerium impurities, in which the resultant Hamiltonian interaction differs qualitatively from the conventional *s*−*f* exchange interaction [30]. According to this new interaction, they worked out the Kondo effect, the spin-disorder resistivity, the Ruderman–Kittel interaction, and the depression of the super-conducting transition temperature are a function of impurity concentration for alloys containing cerium impurities.

*2.1. Exchange Interaction*

Generally, there are three kinds of exchange interaction, including Ruderman–Kittel–Kasuya–Yosida (RKKY) exchange, double exchange, and superexchange.

### 2.1.1. RKKY Exchange

RKKY exchange, referring to the coupling mechanism of nuclear magnetic moments of localized inner *d*- or *f*-shell electron spins in metal through the interaction of conduction electrons, was originally proposed by Malvin Ruderman and Charles Kittel [31]. According to second-order perturbation theory, Ruderman and Kittel put forward a particular indirect exchange coupling, in which the nuclear spin of two atoms interacts with the same conduction electron through the hyperfine interaction, generating correlation energy between these two nuclear spins. RKKY exchange was initially used to explain the broad nuclear spin resonance lines observed in natural metallic silver and one of the most significant applications of RKKY exchange is to predict the existence of giant magnetoresistance (GMR) [32,33]. The computational study of Hussein et al. discovered that RKKY interaction will bring about periodic long-range magnetic interactions in certain doped semiconductor oxides, which are of great significance for the study of spintronic materials [34]. Some researchers confirmed that the magnetic properties of graphene are closely related to RKKY exchange interaction (Figure 1a) [35].

### 2.1.2. Double Exchange

Clarence Zener first proposed the double exchange mechanism, which is a kind of magnetic exchange that may occur between ions in different oxidation states [36]. P. G. de Gennes systematically studied the effects of double exchange in magnetic crystals and drew the conclusion that the special form of double exchange coupling was such that all antiferromagnetic (and also all ferrimagnetic) spin arrangements were distorted as soon as some Zener carriers were present [37]. According to the double exchange theory, it is possible to work out the exchanged single electron between two substances with relative ease and make a prediction for the magnetic properties of certain kinds of materials. The work of Kubo et al. investigated metallic double exchange ferromagnets, such as doped $LaMnO_3$, through an effective Hamiltonian derived from the *s-d* model Hamiltonian. Korotin et al. discovered a self-doped double exchange ferromagnet $CrO_2$ through band structure calculation. They found that $CrO_2$ possesses both localized and itinerant (delocalized) *d* electrons concurrently, resulting in ferromagnetic ordering due to double exchange, which is similar to colossal magnetoresistance (CMR) manganates (Figure 1b) [38]. Ghosh et al. studied the dc magnetization of the critical phenomena in a high-quality single crystal of the double exchange ferromagnet $La_{0.7}Sr_{0.3}MnO_3$, from which they found that the magnetization-field-temperature ($M - H - T$) behavior below and above $T_C$ obeys a certain scaling, following a single equation of state in which $M/(1 - T/T_C)^{\beta}$ is uniquely related to $H/(1 - T/T_C)^{\beta + \gamma}$ [39].

### 2.1.3. Superexchange

Superexchange, also known as Kramers–Anderson superexchange, was first advanced by Hendrik Kramers [40] and later refined by P. W. Anderson [41], and is the antiferromagnetic coupling between two adjacent cations with a non-magnetic cation as the medium. The difference between superexchange and double exchange is the occupancy of the *d*-shell of the two metal ions in the same or differs by two, while the electrons are

itinerant (delocalized) in double exchange. According to superexchange theory, when the two adjacent cations are orientated at 90 degrees to the bridging non-magnetic anion, such an interaction may be ferromagnetic. Junjiro Kanamori made a profound investigation of the relation between the symmetry of electron orbitals and superexchange interaction. His work revealed that if the cation was subjected to a crystalline field arising from octahedrally or tetrahedrally surrounding anions, the sign of the superexchange interaction was closely connected with the cation orbital state; furthermore, the sign of the superexchange interaction was determined from symmetry relations in certain circumstances [42]. A later work of P. W. Anderson examined the theory of indirect exchange in poor conductors from another viewpoint, in which the $d$ (or $f$) shell electrons placed in wave functions are assumed to be exact solutions of the problem of a single $d$-electron in the presence of the full diamagnetic lattice. He came to the conclusion that $d$-electron interactions lead to three spin-dependent effects which, in the usual order of their sizes, were superexchange (always antiferromagnetic), direct exchange (ferromagnetic), and an indirect polarization effect analogous to nuclear indirect exchange, among which superexchange itself is closely related to the poor conductivity, in agreement with experiments [43]. M. A. Gilleo made a thorough study of superexchange interaction in ferrimagnetic garnets and spinels that contained randomly incomplete linkages. He revealed that a magnetic ion interacts with at least two other magnetic ions in different coordinations, because ferrimagnetism is a cooperative phenomenon [44]. Tôru Moriya developed a theory of anisotropic superexchange interaction by extending the Anderson theory of superexchange to include spin-orbit coupling [45,46]. This work derived the antisymmetric spin coupling predicted by Dzialoshinski from purely symmetry grounds and proposed a symmetric pseudo bipolar interaction and a spin arrangement of $CuCl_2 \cdot 2H_2O$ that was different from what was previously expected. In particular, the energy gap between the parallel-spin state and the anti-parallel-spin state was called the exchange energy [47]. Exchange interaction is of great significance in spintronics, magnetism, and other fields, and many frontier applications are derived from it. Actually, in the early 2000s, Divincenzo et al. developed a universal quantum computation method based on exchange interaction. They provided an explicit scheme in which the Heisenberg interaction alone suffices to implement exactly any quantum computer circuit, offering an alternative route to the implementation of quantum computation [48].

### 2.2. Magnetic Anisotropy and Magnetic Domain

Although exchange interaction keeps the spins consistent, it does not align the spins in a specific direction; instead, the spins in a magnet randomly change direction in response to thermal fluctuations. Magnetic anisotropy is a prerequisite for macroscopic magnetic phenomena, and without it, a magnet is superparamagnetic. Magnetic anisotropy originates from several aspects, such as magnetocrystalline anisotropy, shape anisotropy, magnetoelastic anisotropy, and exchange anisotropy. Significantly, ferromagnets usually exhibit an unmagnetized state macroscopically, which is because bulk ferromagnetic material is composed of many tiny magnetic domains (Weiss domains) [26]. Although the spins are aligned within each magnetic domain, if the bulk ferromagnetic material is in the lowest energy state (unmagnetized state), no net large-scale magnetic field exists because the magnetic fields of the different, tiny magnetic domains counteract each other. For a kind of material, if all the magnetic ions in its unit cell point in the same direction, it is ferromagnetic. Otherwise, if the orientation of different magnetic ions is opposing, such material is antiferromagnetic. Additionally, in the latter case, supposing the effect of different magnetic ions with opposite directions cannot be offset, the material will exhibit ferrimagnetism.

### 2.3. Curie Temperature and Néel Temperature

Temperature is another important factor affecting magnetism. The increase of temperature will lead to the enhancements of thermal motion and increase of entropy

increases, which will greatly affect the magnetic properties of the material. When the temperature rises beyond a critical value, a second-order phase transition will occur, and the spontaneity of the whole system will no longer exist, which means the material is completely paramagnetic [49]. In ferromagnetic and antiferromagnetic materials, such values are defined as the Curie temperature and Néel temperature, respectively. The Curie temperature is a critical temperature for ferromagnetism, and determines the highest temperature at which ferromagnetism exists stably. Curie temperature is extremely vital for the application of ferromagnetic materials. Similarly, the Néel temperature is the highest temperature that the antiferromagnetic phase can exist stably; otherwise, the antiferromagnetic phase will transform to a paramagnetic phase when the temperature is higher than the Néel temperature.

### 2.4. 2D ferromagnetic Materials

2D ferromagnetic materials are ideal candidates for next-generation spintronics devices, requiring Curie temperatures far higher than room temperature, which has become the biggest obstacle to its development. Recent theoretical calculations predict that the ferromagnetic coupling of 2D ferromagnetic semiconductors can be significantly enhanced without introducing carriers by isovalent alloying (Figure 1c) [50]. Searching for an effective way to raise the Curie temperature of 2D ferromagnetic materials above ambient temperature is one of the great challenges in this field.

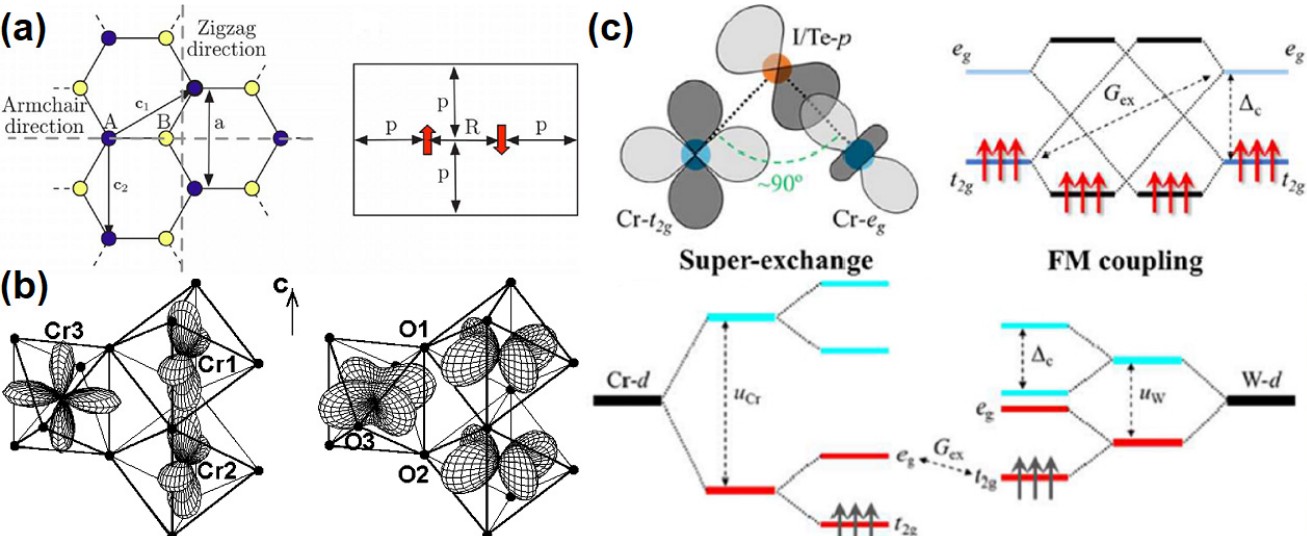

**Figure 1.** (Color online). (**a**) Graphene honeycomb lattice with the two sublattices, A and B, in dark and light colors, respectively. The lattice unit vectors, $c_1$ and $c_2$, lattice constant a = 2.46 Å, as well as the two most common directions, zigzag and armchair, are displayed (left). Unit–cell setups in the antiferromagnetic configuration with the distance, R, between impurity spins are shown, as well as the padding, p, surrounding them (right). Reused with permission from [35]. (**b**) Angular distribution of ab (left) and (yz + zx) (right) electron spin density for the nearest Cr neighbors as determined by LSDA + U. The crystallographic c-axis is indicated. The left panel shows orbitals containing localized spins, and the right panel shows the more dispersive d orbitals forming bands that cross the Fermi level. Reused with permission from [38]. (**c**) Schematic diagrams of the superexchange interaction and ferromagnetic coupling in CrI$_3$ and CrGeTe$_3$ monolayers (upper panel). $G_{ex}$ is the virtual exchange gap and $\Delta_c$ represents the energy gap opened by the crystal field. Red arrows represent the spin-up electrons. Schematic diagrams of orbital evolution and the origin of energy level staggering in Cr and W alloy systems (lower panel). The red and cyan bars represent spin-up and spin-down orbitals, respectively. The $u_{Cr}$ and $u_W$ are the exchange splits for the Cr- and W-d orbitals, respectively. The $\Delta_c$ is the energy gap opened by the crystal field. Reused with permission from [50].

## 3. Ferromagnetic 2D vdW Materials

Ferromagnetic materials possess intrinsic magnetic moments and various exchange interactions, which results in intrinsic moments coupling in the same direction and forming long-range magnetic orders. Under an external magnetic field, the internal magnetization is therefore strong and can be easily observed macroscopically. In fact, ferromagnetism is essentially a form of magnetic sequence [51]. The existence of various defects and impurities in materials will inevitably hinder the movement of domain walls to a certain extent. Consequently, there is a hysteresis phenomenon in the relationship between magnetization intensity and magnetic field in a ferromagnet, which is regarded as a feature of material ferromagnetism.

### 3.1. Intrinsic Ferromagnetic 2D vdW Materials

#### 3.1.1. $Cr_2Ge_2Te_6$

$Cr_2Ge_2Te_6$ crystal was first synthesized by a solid-state reaction method [52]. Bulk $Cr_2Ge_2Te_6$ is a typical semiconductor ferromagnet with a bandgap of 0.7 eV and a Curie temperature of 61 K, in which the atomic layers are stacked in the form of ABC along the *c*-axis, while the chromium ions are located at the center of micro-distorted octahedron comprised of tellurium atom (Figure 2a) [53,54]. To detect the magnetism of conducting ferromagnets, magnetoresistance and the abnormal Hall effect (AHE) are commonly adopted methods, especially for micro/nano devices [55,56]. However, as insulating ferromagnets, such transport measurement cannot be directly applied to $Cr_2Ge_2Te_6$. Lohmann et al. explored an alternative way to probe the magnetic properties of thin $Cr_2Ge_2Te_6$ flakes [57]. They exfoliated $Cr_2Ge_2Te_6$ and constructed a heterojunction between thin $Cr_2Ge_2Te_6$ flakes and Pt. Through this new approach, the detected hysteresis loop of AHE was kept at 60 K, which is close to the Curie temperature of its bulk magnetization. Further density functional theory calculations (DFT) revealed that the reason for the observed AHE was induced ferromagnetism in the Pt part.

Moreover, researchers developed an effective way to construct high-quality $CrSe_2/WSe_2$ vdW heterostructures with clearly resolved Moiré superlattices, in which the dangling-bond-free $WSe_2$ functions as an excellent substrate for the vdW epitaxial growth of ultrathin $CrSe_2$ nanosheets with tunable thickness down to the monolayer limit. These $CrSe_2/WSe_2$ vdW heterostructures display strong air stabilities with robust thickness-dependent magnetic properties [58]. Meng et al. developed a Chemical Vapor Deposition (CVD) strategy to synthesize single crystals of ferromagnetic vdW 1T-$CrTe_2$ with various thicknesses, which provides an ideal platform for the investigation of intriguing physical phenomena and offers the possibility of fabricating high-temperature magnetoelectric devices in the 2D limit because of their layer-dependent magnetic anisotropy and anomalous thickness-dependent $T_C$. 1T-$CrTe_2$, meanwhile, exhibits excellent processing ability in the air and a robust AHE without any encapsulation for samples with a thickness greater than 5 nm, which proves the relatively good stability of this material and demonstrates its potential in future spintronic devices [59]. Moreover, Zhang et al. successfully synthesized ultrathin $CrTe_2$ films via the molecular beam epitaxy (MBE) method. For few-layer films, $CrTe_2$ exhibits intrinsic ferromagnetism with a Curie temperature up to 300 K, an atomic magnetic moment of ~0.21 μB/Cr, and a perpendicular magnetic anisotropy (PMA) constant (Ku) of $4.89 \times 10^5$ erg/$cm^3$ at room temperature, which was unambiguously evidenced by a superconducting quantum interference device and X-ray magnetic circular dichroism. Even in a monolayer sample, ferromagnetic order still exists with $T_C$ ~ 200 K [60].

The $T_C$ of $Cr_2Ge_2Te_6$ is sensitive to thickness, which suggests the interlayer coupling is vital for the ferromagnetic order of $Cr_2Ge_2Te_6$ crystal. Wang et al. fabricated a device based on a few-layered $Cr_2Ge_2Te_6$ with h-BN encapsulation on a silica substrate, in which they observed a strong field effect through electrostatic gating [61]. Low-temperature microregion Kerr measurement of $Cr_2Ge_2Te_6$ transistors was also carried out using an ionic



liquid and solid silicon gate. Additionally, the authors realized a tunable magnetization circuit with different gate doping conditions. Under the different gate doping conditions, the microregion Kerr measurement of the device displayed a bipolar tunable magnetization loop below the Curie temperature, which was mainly caused by torque rebalancing in the spin-polarized band structure.

Tuning the internal electron spin and system magnetism under an electric field has always been a great challenge, and is of great significance for controlling quantum states via electrostatic gating technology for nanoelectronics systems [62]. Recently, Ren et al. theoretically investigated the variations of electronic structures and magnetic properties of bilayer $Cr_2Ge_2Te_6$ with different strains and electrostatic doping concentrations. According to their work, the ferromagnetic stability of bilayer $Cr_2Ge_2Te_6$ can be enhanced within a critical strain range of −3% to 1%, as well as a critical doping range of 0 to 0.2 e/u.c., in theory. Beyond these critical ranges, due to the competition between various exchange interactions, the tensile strain will induce a phase transition from ferromagnetism to antiferromagnetism. Moreover, compressive strain or electrostatic doping induces the magnetization direction to change from out-of-plane to in-plane [63].

### 3.1.2. $Fe_3GeTe_2$, $Fe_5GeTe_2$

$Fe_3GeTe_2$ (FGT) is a kind of vdW magnet, which was first prepared by a direct solid-state reaction [64]. In each FGT monolayer, heterogeneous $Fe_3Ge$ metal plates were sandwiched between two tellurium layers (Figure 2b), with two inequivalent Fe sites, $Fe_I^{3+}$ and $Fe_{II}^{2+}$, per formula unit [64,65]. Deng et al. successfully synthesized high-quality bulk single crystals via the chemical vapor transport (CVT) method and investigated their layer-dependent ferromagnetism under a voltage gate. As the number of layers was reduced to a single layer, although FGT became more insulating, ferromagnetism still existed. However, monolayer FGT lost its strong 2D ferromagnetism and vertical anisotropy, as its Curie temperature dropped rapidly from 207 K to 130 K [65]. The work of Yi et al. revealed that the antiferromagnetic transition of single-crystal $Fe_3GeTe_2$ occurs at 152 K and 214 K, where the antiferromagnetic state is highly anisotropic and the magnetic moment is only parallel to the *c*-axis [66]. The discovery of tunable ferromagnetism and changes in magnetic anisotropy in $Fe_3GeTe_2$ offers an exciting platform for exploring the application of vdW magnets in room-temperature spin electronic devices.

Mechanical exfoliation is an efficient way to obtain few-layer FGT, where $SiO_2/Si$ [67] or gold films are evaporated on $SiO_2/Si$ [68] as substrates. Deng et al. developed an $Al_2O_3$-assisted exfoliation method, which can be used to obtain monolayer FGT [65]. Li et al. prepared FGT crystal by CVT on Si (111) substrate, of which the long-range magnetic order possesses an unconventional out-of-plane stripe domain phase. In the focused ion beam-patterned $Fe_3GeTe_2$ microstructure, the out-of-plane stripe domain phase changed dramatically at 230 K to an in-plane vortex phase that lasted longer than that at room temperature (Figure 2c) [69]. Fei et al. observed an out-of-plane magnetic sequence by polar reflection circular dichroism (RMCD) and magneto-optical Kerr effect (MOKE) microscopy in few-layer FGT, which was fabricated on gold substrates [67]. The decrease of $T_C$ and out-of-plane anisotropy, as well as the transformation from β to 2D Ising, indicates that the FGT monolayer is a real 2D flow ferromagnet (Figure 2d). Moreover, Tan et al. experimentally investigated the anomalous Hall effect on single crystalline FGT nanoflakes and showed that their magnetic properties are highly dependent on thickness. Importantly, a hard magnetic phase with large coercivity and a near-square-shaped hysteresis loop can be observed when the thickness of FGT is less than 200 nm. These characteristics were accompanied by strong PMA, making vdW FGT a ferromagnetic metal suitable for vdW heterostructure-based spintronics [55]. May et al. successfully synthesized $Fe_{5-x}GeTe_2$ bulk single crystals using iodine as a mineralizer/transport agent together with elemental Fe granules, Ge pieces, and Te shot in sealed, evacuated silica ampules that were then exfoliated [70]. Their measurement of the $Fe_{5-x}GeTe_2$ bulk single crystal exhibited a high $T_C$ of 310 K (Figure 2e,f). Significantly, ferromagnetic behavior still exists near

room temperature in flakes exfoliated to a four-unit cell thickness (12 nm). According to the structure and magnetic analysis of $Fe_{5-x}GeTe_2$, a solid solution $Fe_{5-\delta-x}Ni_xGeTe_2$ with x = 1.3 was synthesized [71]. Their results suggested that the substitution of nickel had little influence on its structure. Compared with $Fe_3GeTe_2$, although the application of $Fe_{5-\delta-x}Ni_xGeTe_2$ in spin electronic devices showed similar performance, the adoption of Ni doping instead of using $T_C$ to adjust $\mu_{sat}$ may be advantageous.

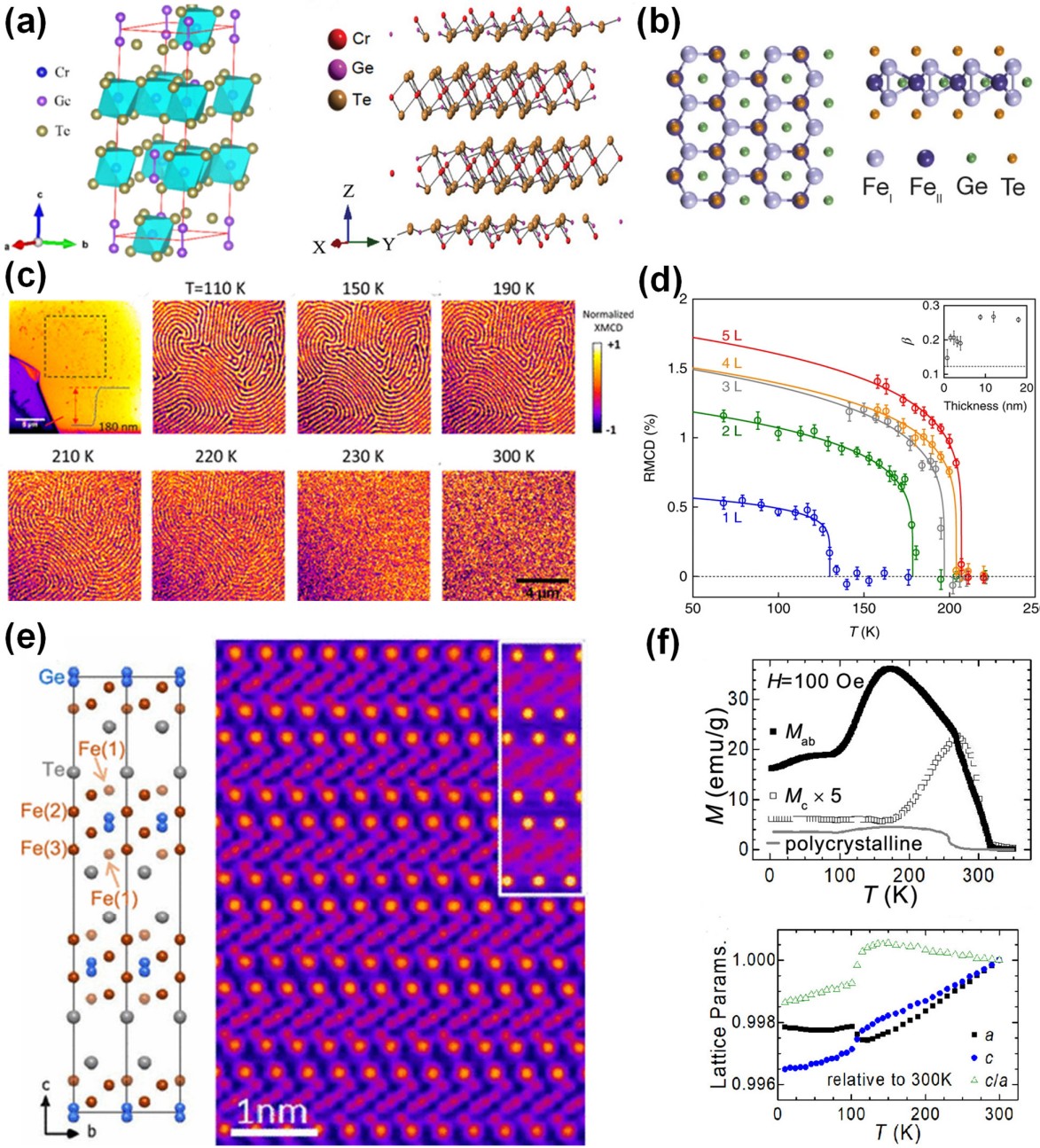

**Figure 2.** (**a**) Crystal structure of $Cr_2Ge_2Te_6$. Reused with permission from [53, 54]. (**b**) Structure of monolayer FGT. The left panel shows the view along [001]; the right panel shows the view along [010]. Reused with permission from [65]. (**c**) Temperature-dependent domain imaging of $Fe_3GeTe_2$. Reused with permission from [69]. (**d**) Criticality analysis for FGT flakes of different thicknesses. Reused with permission from [67]. (**e**) Crystal structure and real space STEM image of $Fe_{5-x}GeTe_2$. The left panel is the average crystal structure obtained from single-crystal X-ray diffraction data

where Fe (1) and Ge are split sites; the right panel is a Z-contrast HAADF image with simulation overlaid in the region outlined by a white box. Reused with permission from [70]. (**f**) Magnetic characterization of bulk single crystals and polycrystalline samples. Temperature dependence of the magnetization upon cooling in an applied field H//c (Mc) and H⊥c (Mab) for single crystals as well as data for the polycrystalline sample (upper panel). Normalized lattice parameters reveal a strong magnetoelastic coupling near 100 K (lower panel). Reused with permission from [70].

### 3.1.3. $CrX_3$(X = I, Br)

Different from traditional metal monolayers, the coupling between magnet and substrate is very weak in $CrI_3$ with $T_C$ = 45 K, which is considered an insulating magnet. Huang et al. exfoliated few-layer $CrI_3$ flakes with graphite encapsulation from a bulk crystal synthesized by mixing chromium powder and anhydrous iodine beads [15]. Their work revealed an interesting layer-dependent ferromagnetic phase in $CrI_3$ flakes, which displayed a transition from monolayer ferromagnetism to bilayer antiferromagnetism, and even back to the ferromagnetism of the trilayer or bulk. Monolayer $CrI_3$, with intrinsic ferromagnetism and layer-dependent magnetic behavior, provides a great platform for studying quantum phenomena, as well as engineering new low-temperature magneto-optical devices.

$CrI_3$ is a typical Ising model ferromagnet with strong out-of-plane anisotropy [72] that has a rhomboid structure [73] stacked in the form of ABC. In every single layer, chromium atoms bond with six I atoms in octahedral coordination to form a honeycomb structure (Figure 3a) [74]. Jiang et al. fabricated dual-gate, field-effect devices of atomically thin $CrI_3$, in which h-BN and few-layer graphene were applied as a gate dielectric and contact electrode, respectively [75]. According to their experimental results, the magnetic properties of mono- and bilayer $CrI_3$ were controlled by electrostatic doping of vertical $CrI_3$ graphene heterostructure. Notably, for bilayer $CrI_3$, electron doping above ~2.5 × $10^{13}$ $cm^{-2}$ induces a ground state transition from antiferromagnetic to ferromagnetic in the absence of a magnetic field (Figure 3b). As reported, $CrBr_3$ is the better choice for magnetic applications compared with $CrI_3$ because of the former's higher air stability [76]. Zhang et al. observed the ferromagnetism of few-layer $CrBr_3$ by polarization-resolved magneto photoluminescence. The light absorption spectrum of $CrBr_3$ exhibits a dependence on the out-of-plane magnetic field, which indicates its potential application in the field of optoelectronics. Unlike bilayer $CrI_3$, with antiferromagnetic coupling between layers in the ground state, six-layer $CrBr_3$ remains ferromagnetic, which indicates its interlayer coupling is ferromagnetic (Figure 3c). Chen et al. successfully grew monolayer and bilayer $CrBr_3$ on highly oriented pyrolytic graphite (HOPG) substrates with the MBE method, in which the direct observation of vdW stacking-dependent interlayer magnetism was realized for the first time [77].

### 3.1.4. $MSe_2$ (M = Mn, V)

In previous studies, massive $MnSe_x$ compounds, such as $MnSe_2$ (pyrite structure) and $\alpha$-MnSe (halite structure), were found not to be ferromagnetic [78,79]. However, J. O'Hara et al. observed room-temperature ferromagnetism in $MnSe_x$ films grown by MBE [80]. They attributed the observed magnetic signal to the intrinsic ferromagnetism of monolayer $MnSe_2$. For thicker crystals, it was speculated that intrinsic ferromagnetism may come from interface magnetism combined with vdW-$MnSe_2$ and/or $\alpha$-MnSe (111). The side view of the lattice clearly showed a single layer of $\alpha$-MnSe (111); that is, the three layers of Se-Mn-Se atoms highlighted by the dotted line in Figure 3d are equivalent to 1T-$MnSe_2$. DFT calculations suggested that for the monolayer $MnSe_2$, its 1T structure was thermodynamically stable and possessed ferromagnetism close to room temperature [81,82].

For monolayer $VSe_2$, in addition to its ferromagnetic properties, the existence of an enhanced charge density wave (CDW) transition temperature, metal-insulator transition, and valley polarization has been authenticated or predicted [83–85]. CVT [86], CVD [87],

and electron beam evaporation [88] have been applied to the synthesis of VSe$_2$ thin films. Recently, Xie's team successfully fabricated ultra-thin (4-8-layer) 2D VSe$_2$ nanosheets by exfoliating large VSe$_2$ crystals in a formamide solvent, in which the electronic and magnetic properties of the 3D bulk phase were well retained [89]. Chen et al. performed DFT studies on the geometric structure, stability, and magnetic and electronic properties of three-dimensional (3D) bulk VSe$_2$, 2D few-layer to monolayer nanosheets, as well as one-dimensional (1D) nanoribbons and nanotubes derived from VSe$_2$ in detail [90]. Bonilla et al. adopted the electron beam evaporation method to prepare few-layer VSe$_2$ on HOPG and MoS$_2$ substrates by simultaneously depositing selenium atoms. The films were annealed at 300 °C for 5 h, in which a strong ferromagnetic order with a $T_C$ higher than room temperature was observed [88]. The work of Yu et al. also verified this result, where monolayer 1T-VSe$_2$ flakes were synthesized by electrochemical exfoliation with organic cations as the intercalants, which were immediately passivated in situ in the solution (Figure 3e) [91]. The magnetization saturation (MS) of VSe$_2$ exhibits a strong temperature dependence, of which $T_C$ reached 470 K. Conspicuous X-ray magnetic circular dichroism (XMCD) signals ($\mu^+$–$\mu^-$ = nonzero) in both their bare and passivated sample confirmed the room-temperature ferromagnetism of VSe$_2$ (Figure 3f). Monolayer VSe$_2$ passivated by mercaptan presented the strongest ferromagnetism, from which it can be deduced that its intrinsic ferromagnetism probably provides a lower limit to the ferromagnetism, while the defect concentration increased the upper limit.

### 3.1.5. MnSn

In early theoretical studies, bulk MnSn in a zinc-blende structure was predicted to be a half-metallic ferromagnet, and was anticipated to be valuable in spintronics owing to their lattice compatibility with semiconductors [92,93]. Yuan et al. successfully synthesized ultrathin MnSn films from monolayer to a few layers by MBE, in which clear magnetic behavior was observed experimentally [94]. Using a superconducting quantum interference device (SQUID) magnetometer (Quantum Design) configured with a longitudinal pick-up coil, the magnetic behavior of the MnSn films showed a clear thickness dependence, which provides a strategy for precisely tuning its magnetism, a new platform for studying 2D magnetism, and the possibility of integrating 2D ferromagnetism with silicon technology. In another work, Li et al. synthesized Fe-doped SnS$_2$ bulk crystals via a direct vapor-phase method and subsequently obtained monolayer Fe-doped SnS$_2$ through mechanical exfoliation. They found that the monolayer Fe-doped SnS$_2$ exhibits ferromagnetic behavior with perpendicular anisotropy at 2 K and a Curie temperature of ~31 K.

### 3.1.6. FeB$_2$ (B = Se, Te)

In a previous study, Liu et al. developed a hydrothermal co-reduction method to fabricate ferroselite (FeSe$_2$) nanorods and frohbergite (FeTe$_2$) nanocrystallites using N$_2$H$_4$·H$_2$O as a reductant. They observed distinct magnetic hysteresis loops of FeSe$_2$ and FeTe$_2$ with a vibrating sample magnetometer (VSM) [95]. Very recently, the sublimed-salt-assisted CVD method at atmospheric pressure (APCVD) was adopted to synthesize layered FeSe$_2$ nanocrystals, which exhibited a phase transition at ≈11 K [96]. Recently, Kang et al. successfully realized the phase-controllable growth of ultrathin 2D magnetic FeTe crystal. According to their work, antiferromagnetic tetragonal FeTe had a Néel temperature that gradually decreased from 70 to 45 K as the thickness declined from 32 to 5 nm, and the ferromagnetic hexagonal FeTe was accompanied by a drop of Curie temperature from 220 K (30 nm) to 170 K (4 nm) [97]. Meanwhile, via the CVD method, Wang et al. successfully synthesized ultrathin FeTe nanosheets with tetragonal and hexagonal phases by tuning the growth temperature according to the phase diagram. Their magneto-transport measurements illustrated that the tetragonal crystal displays a linear magneto-resistance (LMR) as high as 10.5% at 1.9 K and the hexagonal nanosheets showed 5.8% LMR at 1.9 K, indicating their potential applications as magnetoresistive devices.

Moreover, tetragonal FeTe illustrated a superconducting transition at 9.0 K, which may arise from oxygen doping, indicating that FeTe is an intriguing material platform for exploring 2D magnetism and superconductivity, and their heterostructures have potential applications in novel functional devices [98].

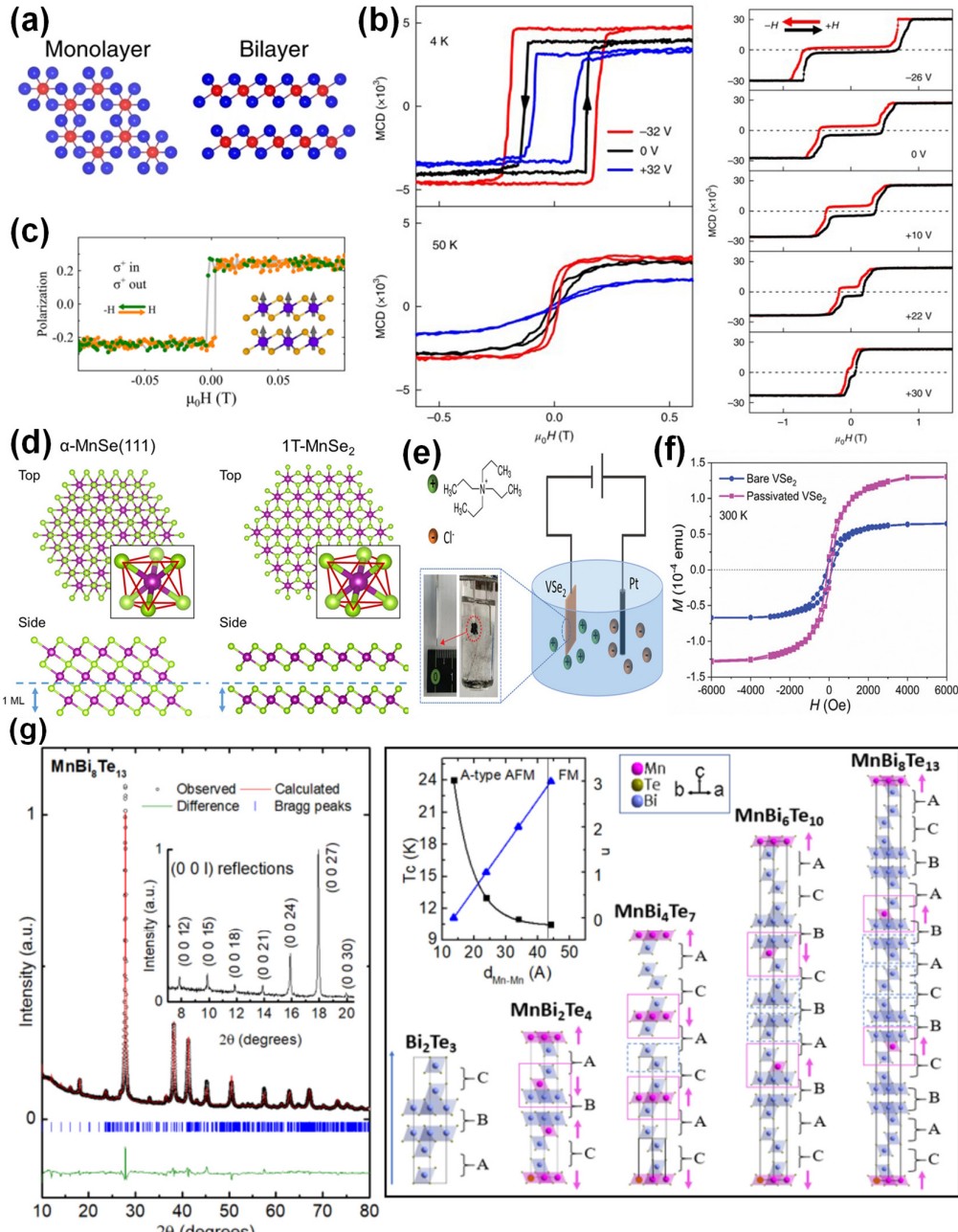

**Figure 3.** (**a**) Top view of monolayer CrI₃, where the Cr atoms (red balls) form a honeycomb structure in edge-sharing octahedral coordination by six I atoms (blue balls), and side view of bilayer CrI₃ of the rhombohedral stacking order. Reused with permission from [74]. (**b**) MCD versus magnetic field (left) at three representative doping levels at 4 K (top panel) and 50 K (bottom panel); MCD versus magnetic field at 4 K at representative gate voltages (right). Reused with permission from [75]. (**c**) Hysteresis loop for bilayer CrBr₃. Reused with permission from [76]. (**d**) Crystal structure diagrams of α-MnSe (111) and monolayer 1T-MnSe₂. Reused with permission from [80]. (**e**) Schematic illustration of the electrochemical setup for cathode exfoliation of VSe₂ using tetra

propylammonium cation (TPA) as the intercalant. Inset: photograph showing a small piece of a $VSe_2$ crystal clamped by two titanium plates (left), and visible expansion of the $VSe_2$ cathode with a fluffy shape (right). Reused with permission from [91]. (**f**) *M–H* hysteresis loop of bare and passivated $VSe_2$ flakes at 300 K, shown in (**e**). Reused with permission from [91]. (**g**) Structure determination and the stacking rule in $MnBi_{2n}Te_{3n+1}$. Reused with permission from [99].

### 3.1.7. $MnBi_8Te_{13}$

Hu et al. grew single-crystal $MnBi_8Te_{13}$ via the self-flux method and obtained their crystal structure using the Rietveld method to refine its powder X-ray diffraction (XRD) pattern at 300 K. They found alternating stacked monolayers of $MnTe_6$ octahedra, well separated from many monolayers of $BiTe_6$ octahedra along the *c*-axis, in the crystal structure of $MnBi_8Te_{13}$ (Figure 3g) [99]. Interaction between topology and magnetism was the key to realizing the new topological state. They confirmed that $MnBi_8Te_{13}$ was ferromagnetic below 10.5 K, with strong coupling between magnetic and charge carriers. First-principles calculation and an angle-resolved optical emission spectrum further proved that $MnBi_8Te_{13}$ was an intrinsic ferromagnetic axion insulator, which means $MnBi_8Te_{13}$ is an ideal system for studying the Quantum Anomalous Hall Effect (QAHE) and quantum topological magnetoelectric effects.

### 3.2. Extrinsic Ferromagnetic 2D vdW Material

TMDs are an important member of the 2D-materials family, and they have been extensively studied over the past decade. However, due to a lack of intrinsic ferromagnetism in most TMDs, their application in spintronics has been hindered. Therefore, ways to realize the intrinsic ferromagnetism of TMDs with high Curie temperature will be one of the challenges in this field in the future. Based on the original crystal structure of TMDs, several existing methods, such as defect engineering, chemical doping, grain boundary, and the magnetic proximity effect, can not only explore the existence of intrinsic ferromagnetism in stable, few-layer 2D materials, but also promote ferromagnetic stability, which may generate magnetic moments in the system.

### 3.2.1. $In_2Se_3$

$In_2Se_3$, as a multi-ferrous substance, is a bizarre kind of material that can retain ferroelectricity and piezoelectricity concurrently, even down to monolayer [100]. Owing to its extraordinary sustainability and performance, $In_2Se_3$ is usually applied in the ferroelectric field, such as for non-volatile memory [101]. $In_2Se_3$ is a 2D layered semiconductor with a bandgap of 1.3 eV at room temperature [102], while monolayer $In_2Se_3$ has five triangular atomic lattices in the Se-In-Se-In-Se lattice sequence connected with a covalent bond, which composes a quintuple layer (QL) (Figure 4a) [103]. Although there are many phases ($\alpha$, $\beta$, $\gamma$, $\delta$, and $\kappa$ phases) of $In_2Se_3$, as determined by different temperatures and the preparation conditions [104–106], only the $\alpha$ and $\beta$ phases are semiconductors, with layered structures combined by vdW interaction and stable at room temperature [107]. Ferroelectric polarization originates from the asymmetric position of the selenium atom in the middle, which spontaneously breaks the central symmetry. Currently, $In_2Se_3$ often forms heterojunctions or dopants with other materials to generate more characteristics; for example, hole doping induces the tunable ferromagnetism of $\alpha/\beta$-$In_2Se_3$ monolayers. Via theoretical simulation, Sun et al. showed that arsenic doping at the position of selenium could enhance ferromagnetism, making $\alpha/\beta$-$In_2Se_3$ a semimetal, which exhibits strong magnetoelectric coupling with the vdW heterostructure of bipolar magnetic semiconductors in 2D $FeI_2/In_2Se_3$ [108].

### 3.2.2. $MoS_2$

$MoS_2$ monolayers show an S-Mo-S sandwich structure with a direct bandgap of 1.8 eV, in which each Mo atom and six S atoms constitute a triangular prism coordination.

The *d* orbital of the Mo atom has a strong spin–orbit interaction, which leads to coupling between the spin degree of freedom and valley degree of freedom [109–111]. Although there is no intrinsic ferromagnetism in $MoS_2$, researchers have explored some feasible methods, such as morphology preparation [112], external strain [113], and impurity doping [114] to introduce magnetism into $MoS_2$. DFT calculation of Jeon et al. suggested a hydrogen atom on a $MoS_2$ nanosheet tends to combine with the sulfur atom, which will produce a 1 μB magnetic moment [115]. Additionally, further research suggested that the adsorption of some nonmetallic elements such as B, C, N, and F will also transform $MoS_2$ from nonmagnetic to magnetic [116].

Doping is considered an effective technology for modulating the electronic and magnetic properties of 2D materials. Recent studies have shown that it is practicable to realize the room-temperature ferromagnetism of $MoS_2$ through transition-metal-doped $MoS_2$ nanostructures [117,118]. Cheng et al. corroborated that $MoS_2$ can be ferromagnetic if doped with Mn, Zn, Cd, and Hg, and antiferromagnetic if doped with Fe and Co when the impurity concentration of the TM atoms is up to 6.25% [119]. However, later studies [120] found that $MoS_2$ doped with Mn-, Fe- and Co was ferromagnetic. Recently, Fan et al. indicated that V-, Mn-, Fe-, Co-, or Cu-doped monolayer $MoS_2$ can produce good diluted magnetic semiconductor at low impurity concentrations, while Co and Cu doping can produce a higher total magnetic moment. In particular, copper doping will generate strong ferromagnetism relative to the local spin (Figure 4b) [121]. However, because of the presence of strong diamagnetic substrates, although large magnetic moments can be obtained at room temperature by chemical methods, it is difficult to characterize the magnetic states of mono- or multi-layer TMD on possible devices. Han et al. investigated electron irradiation, adopting higher acceleration energies than the displacement energy of Mo atoms, which may be an effective and simple method to achieve the conversion from diamagnetic to ferromagnetic at room temperature by inducing a phase transition in the substrate [122].

### 3.2.3. $PtSe_2$

There is a central symmetric spatial group, P3m1, in the global structure and polar point groups, $C_{3v}$ and $D_{3d}$, at the Se and Pt sites in bulk $PtSe_2$ [123]. Monolayer $PtSe_2$ is semiconductive, with a bandgap of 1.2 eV. However, with an increase of the layer numbers, the bandgap of $PtSe_2$ gradually decreases and finally converts into a semimetal [124]. Such a wide layer-dependent bandgap range of $PtSe_2$ enables it an excellent candidate for electronic and photoelectric applications, such as field-effect transistor (FET), gas sensors, photoelectric detection, and energy storage. Yao et al. fabricated $PtSe_2$ flakes, grown by the direct selenization of Pt (111) until the substrate was covered completely [123]. The $PtSe_2$ monolayer consisted of a Pt layer sandwiched between two Se layers, forming a triangular structure when projected onto the (001) plane. DFT calculations showed that the first three valence bands (marked by α, β, and γ) were mainly contributed by the *p* orbits of Se, and the fourth valence band (marked by δ) was mainly contributed by the *d* orbits of Pt (Figure 4c).

$PtSe_2$ is an ideal platform for studying defect-induced magnetization [125]. Avsar et al. obtained $PtSe_2$ flakes by mechanical exfoliation from bulk crystal [17]. Electrical measurement of the $PtSe_2$ crystal indicates that the ferromagnetic or antiferromagnetic ground state order in the ultrathin material was sensitive to the number of layers. To investigate such an effect further, another $PtSe_2$ device was fabricated containing an additional layer of thick debris on the channel's surface, and it was found that the magnetic transport response of the sample was very interesting because ferromagnetism and antiferromagnetism order coexisted (Figure 4d). It can be concluded that the switching fields of these two kinds of comparative magnetic sequences were different, where the strong layer-dependence relationship between different mechanisms could stabilize the magnetic order differently, which led to different switching fields. This result is consistent with that of the coercive field in the parity layers of $CrI_3$ magnetic-insulating materials measured by the

magneto-optical Kerr effect. Theoretical calculation suggests that Se, especially at the Pt vacancy in monolayer PtSe₂, has a great influence on its magnetic and electronic properties, in which the magnetic moment of the single and double Pt vacancies can reach 6 μB [17].

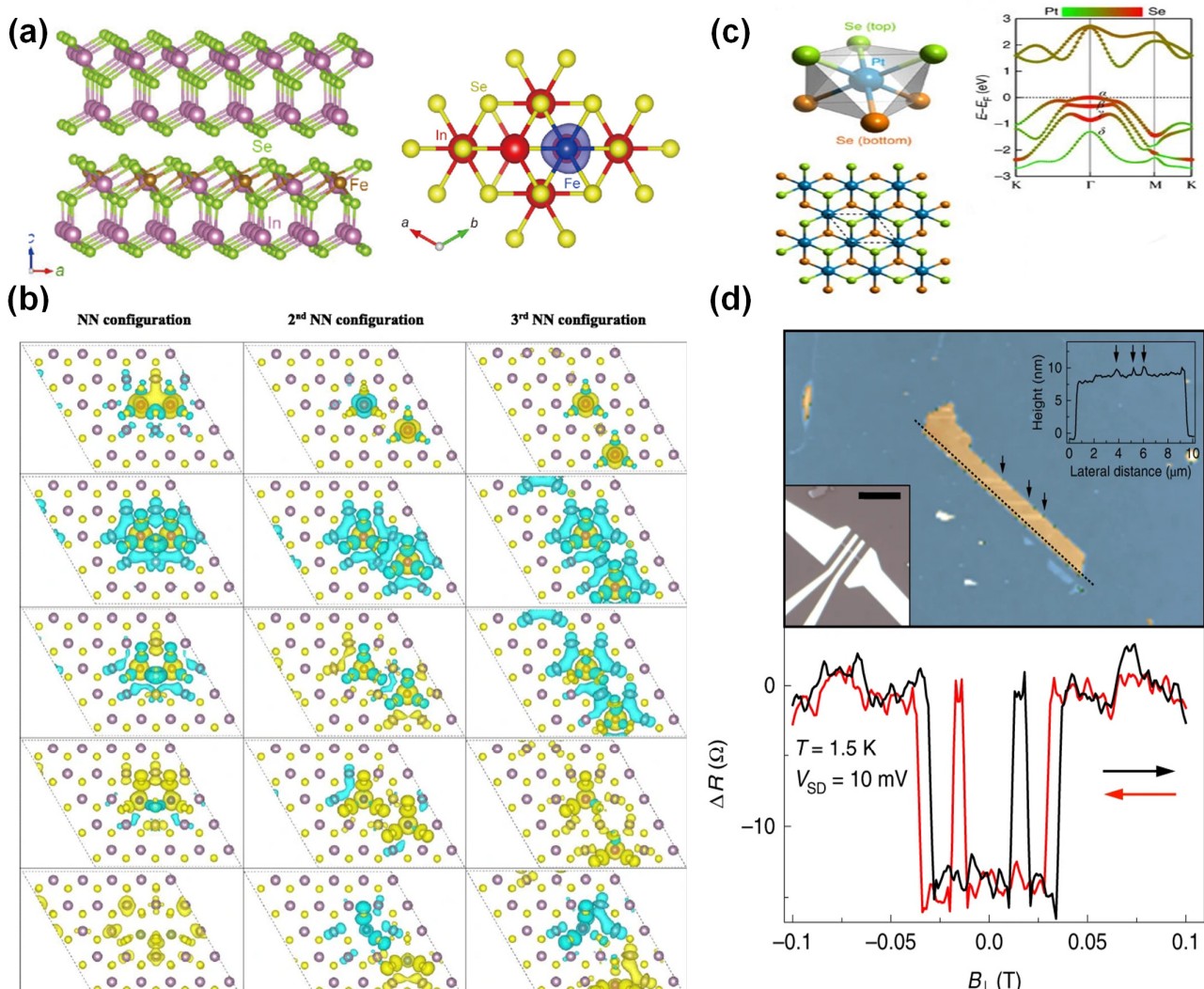

**Figure 4.** (**a**) Crystal structure of the Fe-In₂Se₃ monolayer. The red, yellow, and blue balls represent In, Se, and Fe atoms, respectively. Reused with permission from [103]. (**b**) Spin-resolved charge density isosurface (isosurface value at 0.002 e/Å³) of a TM-doped MoS₂ monolayer at 8% impurity concentration. Reused with permission from [121]. (**c**) Crystal structure of monolayer PtSe₂ (left panel, top: unit cell, bottom: top view); band dispersions along the K-G-M-K direction from first-principles calculations. The color and line width distinguish the contributions from Pt and Se (right panel, red for Se, and green for Pt). Reused with permission from [123]. (**d**) Optical of a completed PtSe₂ device (upper panel). Magnetic-field dependence of ΔR measured by device E (shown in upper panel) at $T = 1.5\,\text{K}$ and $V_{SD} = 10\,\text{mV}$. The red (black) arrow represents the sweep direction from positive (negative) to negative (positive) values. Reused with permission from [17].

### 3.2.4. PdSe₂

Oyedele et al. synthesized and exfoliated monolayer PdSe₂ from bulk crystals with a bandgap of 1.3 eV. The unit cell of PdSe₂ (P21/C space group) contains two palladium atoms and four selenium atoms with lattice parameters a = 5.75 Å (*y*-axis) and b = 5.93Å (*x*-axis) [126]. Notably, the semiconductor PdSe₂ monolayer had remarkable air stability

under ambient conditions. $PdSe_2$ is a typical vdW layered material, in which each Pd atom is connected to four Se atoms in every single layer (Figure 5a) [127]. Zhang et al. theoretically predicted that hole doping in monolayer $PdSe_2$ could induce stoner ferromagnetism, and $T_C$ could be far higher than room temperature, reaching 800 K. Significantly, the hole concentration threshold for ferromagnetism decreases with applied stress (*x*-axis strain) (Figure 5b–e) [128]. For the 2D-materials family, strain engineering is a powerful technology for adjusting their physical quantities and generating new characteristics, such as band gaps, carrier mobility, and exciton absorption [129–131]. In the field of spintronics, magnetic anisotropy, which is defined as magnetization in a specific direction, is an important parameter. In recent years, modulating magnetic properties (such as anisotropy and $T_C$) through strain engineering has become a research topic in magnetic-field regulation. Generally speaking, the 2D atomic structure exhibits unique mechanical advantages, which can theoretically endure larger strain than bulk material.

### 3.2.5. $VTe_2$

Ferromagnetism has already been reported in perfect $VX_2$ (X = S, Se, and Te) monolayers in theory [132]. Some researchers have revealed that the ferromagnetism of 2D $MX_2$ nanostructures can be effectively improved by applying strain and hydrogenation [115, 128]. However, interactions between antiferromagnetic and ferromagnetic ground states have not been achieved in 2D nanostructures. According to first-principles calculation, Pan et al. predicted that a switch of monolayer hydrogenated vanadium telluride ($VTe_2$-H) by applying tension can be realized [133]. They also proved that with an increase of tension, monolayer $VTe_2$-H switches from anti-ferromagnetism to ferromagnetism via paramagnetic turning point accompanying with electronic evolution from semiconductor to metal, further to half-metal. In addition, under the dual influence of magnetic field evolution and increasing tension, its electrical properties could change from semiconductor to metal, and then to half-metal. Its calculated band structure (Figure 5f) indicates that monolayer $VTe_2$-H is a kind of direct band semiconductor under zero tension with a bandgap of 0.48 eV. As the applied tension increases, the bandgap width displays a downward tendency. When the applied tension reaches a critical value of 5%, conduction band minimum (CBM) and valence band maximum (VBM) cross each other at the Fermi level, resulting in metal conductivity of monolayer $VTe_2$-H. Further research proved that antiferromagnetism with semiconductor or metal characteristics at low voltage was caused by superexchange or superexchange enhanced by carriers, while ferromagnetism with semimetal characteristics at high voltage could be generated through double exchange mediated by carriers. Lasek et al. developed an effective way to synthesize ultrathin $VTe_2$ films via MBE [134]. According to their result, monolayer ditellurides with octahedral 1T structure were readily grown, but for multilayers, transition metal dichalcogenide formation competed with self-intercalated compounds.

### 3.2.6. $WS_2$

Monolayer $WS_2$ presents a graphene-like honeycomb structure called H-$WS_2$, which is a triangular prism structure with two S atomic layers in the W atomic layer with the sequence S-W-S, which is a P-6M2 ($D_{3h}$) point group symmetry. Previously, surface-functionalized $WS_2$ nanosheets were synthesized and applied to negative electrodes for lithium-ion batteries [135]. Ma et al. systematically studied the electromagnetic properties of $MoSe_2$, $MoTe_2$, and $WS_2$ monolayers adsorbed by ideal, vacancy-doped, and nonmetallic elements (H, B, C, N, O, and F) through first-principles calculation [136]. Their results confirmed that natural vacancies are not the reason for the generation of $WS_2$ magnetism, but the absorption of hydrogen on $WS_2$, $MoSe_2$, and $MoTe_2$ monolayers, which will bring about a large spatial expansion of spin density and weak antiferromagnetic coupling between local magnetic moments. In previous research, due to the quantum confinement effect, unusual Dirac fermions were observed in graphene [137]. However, simultaneously, some researchers discovered that there is an energy gap in 1D graphene

nanoribbons [138]. The unusual coexistence of Dirac fermions and energy gaps makes graphene zigzag nanoribbons (ZNRs) exhibit a half-integer oscillation state between the semiconductor and metallic state as a function of width [139].

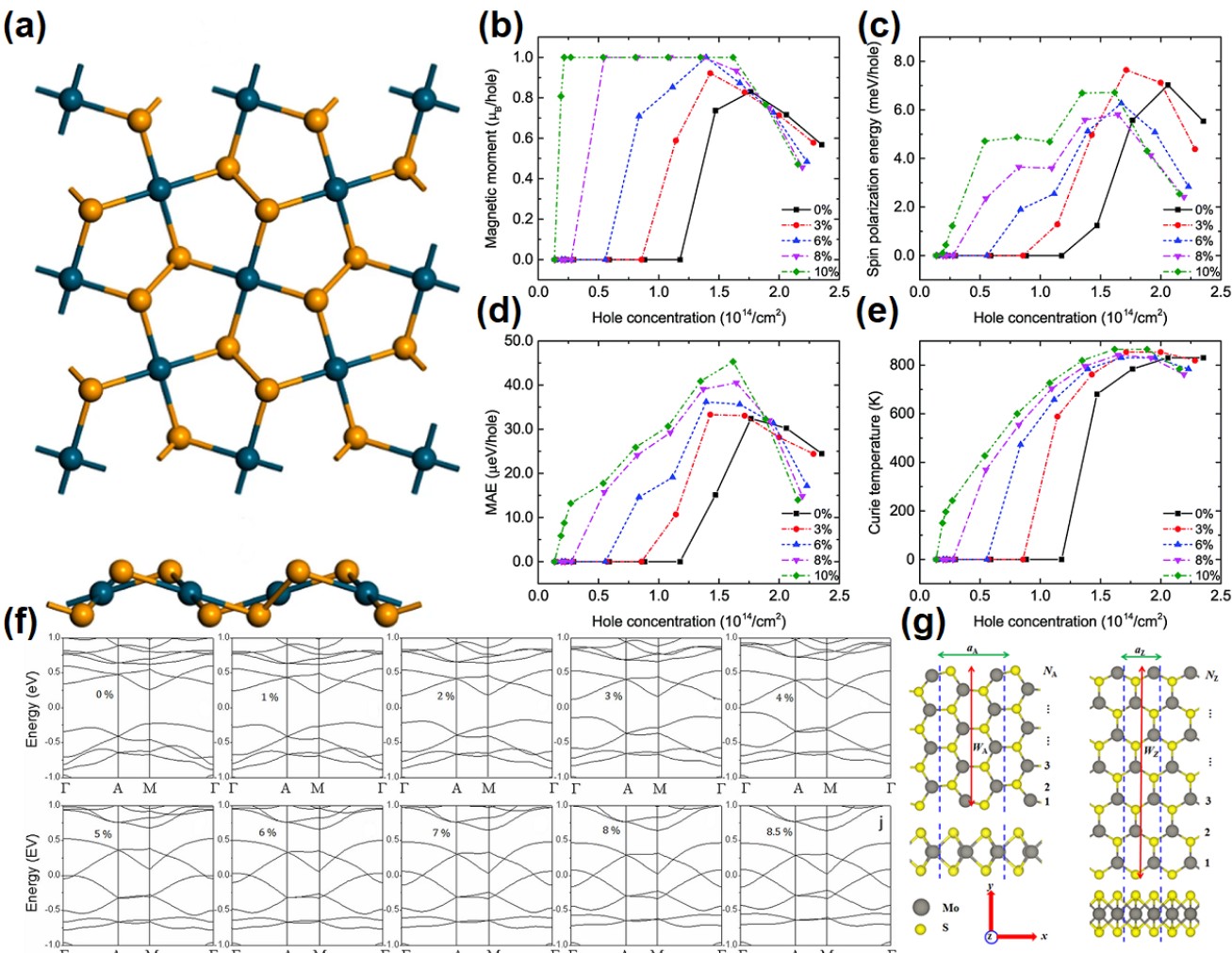

**Figure 5.** (**a**) The top (upper panel) and side (lower panel) views of PdSe₂ monolayer, respectively. Reused with permission from [127]. (**b–e**) Magnetic moment–spin polarization energy, magnetic anisotropic energy (MAE), and Curie temperature versus hole doping concentration of PdSe₂ under *x*-axis strains of 0%, 3%, 6%, 8% and 10%. Reused with permission from [128]. (**f**) Calculated band structures of VS₂-H monolayers under tension ranging from 0 to 18%. Reused with permission from [133]. (**g**) ANRs and ZNRs. The gray (yellow) balls stand for W (S) atoms. The unit cells are depicted by the blue dashed lines. The lattice constants, $a_A$ and $a_Z$, of the ANRs and ZNRs are indexed by green arrows, whereas the widths, WA and WZ, of the ANRs and ZNRs are depicted by red arrows, which are also described by $N_A$ and $N_Z$, respectively. Reused with permission from [140].

Based on this unusual phenomenon of graphene, Zhang et al. successfully synthesized 1D MoS₂ and WS₂ nanoribbons (diameters of 1–4 nm) with templates of carbon nanotubes [140]. They fabricated two kinds of WS₂ nanoribbons, armchair nanoribbons (ANRs) and zigzag nanoribbons (ZNRs) (Figure 5g), by cutting the monolayer WS₂ in different directions. A systematic study was then conducted to explore the atomic structure, electrical properties, and magnetism of 1D WS₂ nanoribbons with armchair and zigzag properties through first-principles calculation. Their results suggested that the characteristics of the two types of nanoribbons were quite different. WS₂ ANRs maintains the non-

magnetic and semiconductor properties of the bulk nanoribbons, while the ZNRs shows ferromagnetic and metallic properties. Additionally, the band gaps of the nanoribbons decreased remarkably because of the existence of zigzag edges. Obviously, the electrical and magnetic properties of $WS_2$ nanoribbons, to a considerable extent, were affected by the type of edge. More importantly, the electrical or magnetic properties of ANRs and ZNRs can be adjusted by applying an external strain. Recently, Zhang et al. successfully developed a universal, scalable, and controllable synthesis route for V-doped $WS_2$ atomic layers as a dilute magnetic semiconductor by CVD, in which the vanadium doping induced inherent ferromagnetic ordering at room temperature, with the strongest ferromagnetic signal for the moderately doped (2 at%.) sample [141].

### 3.2.7. SnS2

Su et al. proposed a novel strategy for the controlled CVD synthesis of thin crystal arrays of layered semiconductors $SnS_2$, in which the few-layer $SnS_2$ was demonstrated to be an ultrasensitive photodetector [142]. In another work, Li et al. synthesized Fe-doped $SnS_2$ bulk crystals via a direct vapor-phase method and subsequently obtained monolayer Fe-doped $SnS_2$ through mechanical exfoliation. According to the VSM measurement carried out via a physical properties measurement system (PPMS), monolayer Fe-doped $SnS_2$ exhibited ferromagnetic behavior, a perpendicular anisotropy at 2 K, and a Curie temperature of ~31 K [143].

### 3.2.8. GaSe and GeSe

In an early study, Zhou et al. successfully realized the controllable synthesis of single-crystalline ultrathin GaSe nanoplates on a flexible mica substrate via vdW epitaxy. Individual 2D GaSe nanoplates were found to show typical *p*-type conductance and good light-response characteristics [144]. Concurrently, Li et al. synthesized ultrathin 2D GaSe via CVT in a tube furnace system, which showed high photoresponse and field-effect transistor (FET) characteristics [145]. The magnetic behavior of GaSe has been reported many times in theoretical studies [146–149]. The 2D unit cell of a GaSe monolayer is composed of two Ga atoms and two Se atoms. The two neighboring Ga atoms are vertically stacked and are sandwiched between two planes of Se atoms. Gao et al. performed first-principles calculations using DFT in the local density approximation (LDA) for the exchange-correlation function as implemented in the QUANTUM ESPRESSO package. Their theoretical work revealed that tunable ferromagnetism and half-metallicity in monolayer GaSe could be induced by hole doping [146]. Soon afterward, Lu et al. systematically investigated the structural and electronic properties of Fe-adsorbed monolayer GaSe using first-principles DFT calculations. Attributed to the transfer of Fe-3d electrons, magnetic moments of −0.004 μB and −0.178 μB were induced for the nearest neighbor Ga and Se atoms, respectively, which showed antiferromagnetic coupling with the adsorbed Fe atom [147]. Similarly, Tang et al. systematically studied the modulation of an electric field on the structural, electronic, and magnetic properties of a vacancy-doped GaSe monolayer using first-principles DFT calculations. According to their work, an unpolarized electronic structure in the V–Se doped GaSe monolayer was seen, whereas a half-metallicity with a 1 μB magnetic moment was seen for the V–Ga-doped system. The stability of the magnetic moment was enhanced with an increase of the vertical electric field along the −*z* direction, but was weakened along the +*z* direction, which led to a prompt switch from a magnetic to the nonmagnetic state under a +0.6 V/Å electric field [148]. Recently, in another computational study, Ke et al. predicted that the magnetic properties of Fe-doped monolayer GaSe could be controlled by an electrical field [149]. In theory, they proved that the spin polarization, magnetic moment, and magnetic anisotropic energy (MAE) strongly depended on the applied vertical electric field arising from a magnetic configuration modulation in the system.

Bulk GeSe single crystal can be synthesized via solid-state reaction in an evacuated silica tube [150]. Xue et al. successfully synthesized micrometer-sized GeSe nanosheets

and investigated their anisotropic photoresponse properties [151]. Recently, Sarkar et al. synthesized rhombohedral $(GeSe)_{0.9}(AgBiSe_2)_{0.1}$ crystal by Bridgman method, which showed ferroelectric-instability-induced, ultralow thermal conductivity, and high *p*-type thermoelectric performance [152]. Early computational research revealed that monolayer GeSe is a semiconductor with a direct bandgap of 1.16 eV with strong stability. Additionally, the electronic band structure of monolayer GeSe and the small effective mass of charge carriers are sensitive to strain [153]. In 2017, Zhao et al. experimentally obtained monolayer GeSe on $SiO_2/Si$ substrates with a controlled laser thinning method in a high vacuum and systematically studied the band structure and photoelectric of the GeSe monolayer [154]. However, investigation of the magnetic properties of GeSe is still lacking. Ishihara et al. reported negative magnetoresistance in GeSe, and they attributed this phenomenon to carrier scattering by localized magnetic moments [155].

## 4. Antiferromagnetic 2D vdW Materials

### 4.1. $NiGa_2S_4$, $FeGa_2S_4$, $Fe_2Ga_2S_5$

$AGa_2S_4$ (A = Ni, Fe) and $Fe_2Ga_2S_5$ are layered sulfide insulators with triangular lattices, and the structure of $AGa_2S_4$ can be described by a plate composed of two GaS layers and one $AS_2$ layer stacked along the *c*-axis and separated by a vdW gap (Figure 6a) [156,157], in which the $AS_2$ layer is the same as the $CoO_2$ layer of the superconductor $Na_x$-$CoO_2 \cdot yH_2O$ [158]. $Fe_2Ga_2S_5$ possesses a bilayer $FeGa_2S_4$ structure, where a $Fe_2S_3$ layer composed of a pair of $Fe^{2+}$ triangular planes replaces the central $FeS_2$ layer of $FeGa_2S_4$. Nakatsuji et al. proved that a high-quality $NiGa_2S_4$ sample was a bulk-insulated antiferromagnet located on a regular triangular lattice, which exhibits spin disorder in two dimensions [159]. According to their study, $NiGa_2S_4$ was a low-spin ($S \leq 1$) quasi-2D bulk magnet on an equilateral triangular lattice, of which the spin disorder could be stabilized by geometric frustration at low temperature. Later, they also reported a single crystal study of $NiGa_2S_4$, $FeGa_2S_4$, and homologous double triangular antiferromagnetic $Fe_2Ga_2S_5$ via isostructural single-layer triangular atomic force microscopy, in which the $NiGa_2S_4$ single crystal was synthesized by CVT with iodine [160]. Strikingly, the magnetic properties of $FeGa_2S_4$ bore strong resemblances to those of $NiGa_2S_4$, despite the fact that $Fe^{2+}$ has an S = 2 spin twice as large as that of S = 1 for $Ni^{2+}$, while both of them have Heisenberg spin and form a frozen disordered state below T ~ $\theta_w$ = 10 K, thus showing a $T^2$ dependence that is similar to that of $NiGa_2S_4$. Such similarities strongly indicate that the coherent 2D behavior has the same mechanism as frozen spin disordered states. However, an antiferromagnetic transition was observed in $Fe_2Ga_2S_5$, from which it can be concluded that the geometric frustration of the single-layer triangular lattice will stabilize the spin disordered state.

### 4.2. $XPS_3$ (X = Fe, Ni, Mn)

Single crystals of $XPS_3$ can be synthesized by CVT and exfoliated to a few-layer, or even a monolayer surface [161–163]. According to earlier findings, $FePS_3$ is an Ising-type antiferromagnet [161], while $NiPS_3$ belongs to the XXZ type [162]. However, research of renormalization groups revealed that the behavior of the antiferromagnetic 2D quantum spin system with easy-plane-like anisotropy is similar to that of the XY system at low temperature [164,165]. The Néel temperature ($T_N$) of $FePS_3$ and $NiPS_3$ is 120 K and 155 K, respectively [166]. Both of these materials have the same basic crystal structure, a C2/M monoclinic cell [167] and a honeycomb metal ion arrangement (Figure 6b) [168]. These metal ions are separated by a wide vdW gap (magnetic potential) and form very close to an ideal hexagon in the a–b plane, which possesses weak bonds and interacts along the vertical *c*-axis. $XPS_3$-type material can be cleanly and easily peeled into a controllable thin layer by the "transparent tape" method, and will have high air and water stability.

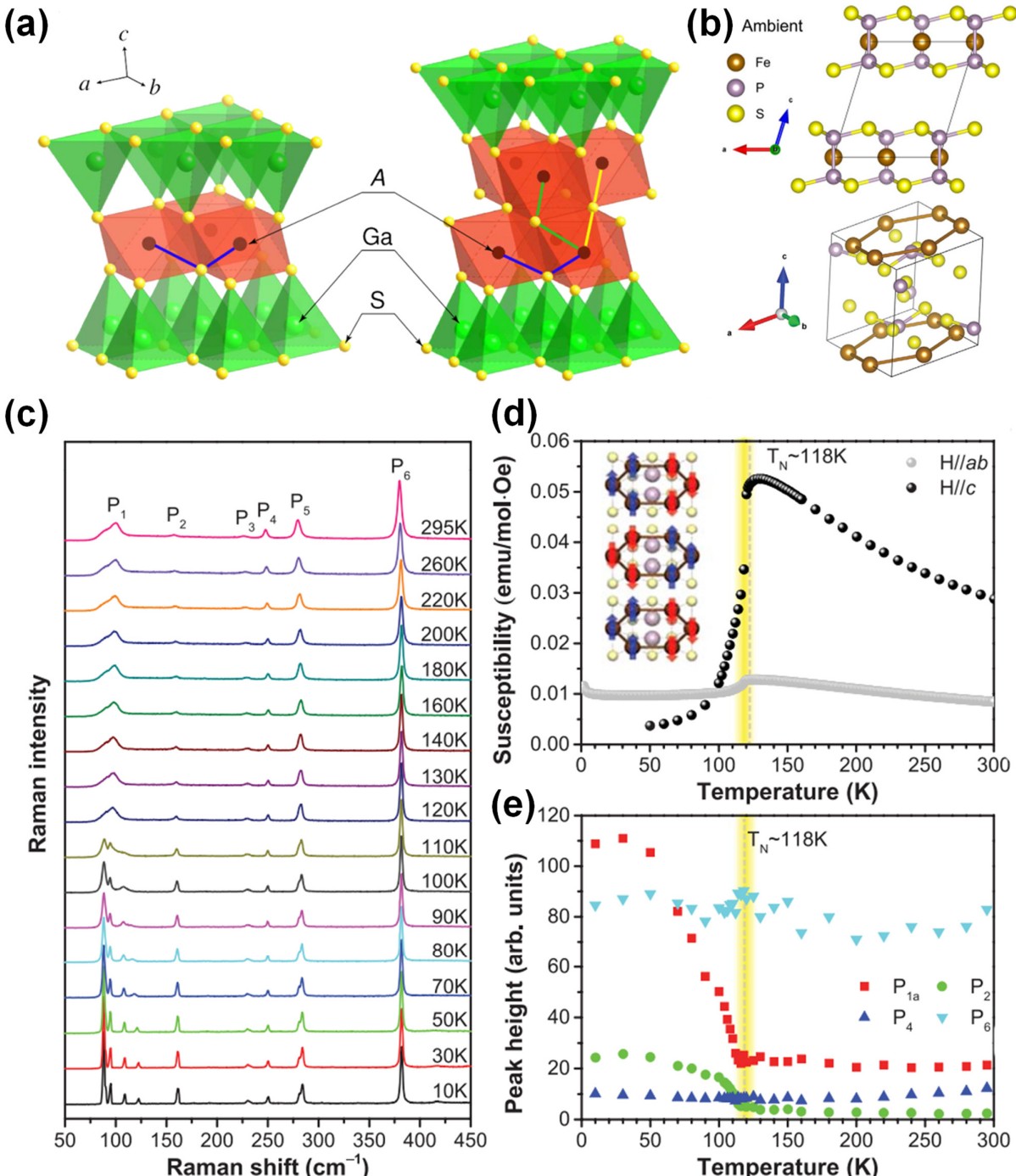

**Figure 6.** (**a**) Structure of the unit slab for single layered AGa₂S₄ (A = Ni, Fe) (Left) and bilayer Fe₂Ga₂S₅ (Right). Superexchange paths are shown by lines within the diagrams (color online). Re-used with permission from [160]. (**b**) Crystal structure of ambient-pressure FePS₃. Reused with permission from [168]. (**c**) Temperature dependence of Raman spectrum for bulk FePS₃. Reused with permission from [161]. (**d**) Temperature dependence of magnetic susceptibility for bulk FePS₃ along with the *a*- and *b*- (black spheres) and *c*- (gray spheres) axes. Reused with permission from [161]. (**e**) Temperature dependence of the intensities of several Raman peaks for bulk FePS₃. Reused with permission from [161].

Neutron scattering is an important tool for characterizing antiferromagnetic materials; however, it is not applicable for 2D vdW materials owing to their very small sizes [169,170]. Kim et al. used Raman spectroscopy to characterize antiferromagnetic order, and observed some changes in the magnetic phase transition characteristics of $FePS_3$ in their Raman spectra [171]. According to the Raman spectra of $FePS_3$ in parallel [*z(xx)z*] and cross [*z(xy)z*] polarization configurations as a function of temperature (Figure 6c), P2 split into two peaks at low temperature, which is the clearest indication of magnetic ordering. Different splitting peaks were also observed concurrently under different polarization structures concurrently. The magnetic susceptibility of $FePS_3$ exhibited obvious antiferromagnetic phase transitions, in which the change of magnetic susceptibility along the *c*-axis was the largest, indicating that $FePS_3$ has an Ising-type magnetic sequence (Figure 6d). The Néel temperature was almost independent of the number of layers, which means that the interaction between layers has little effect on the magnetic ordering in $FePS_3$ (Figure 6e).

### 4.3. $MnBi_{2n}Te_{3n+1}$

$MnBi_2Te_4$, $MnBi_4Te_7$, and $MnBi_6Te_{10}$ belong to the same class of materials, and its chemical formula can be summarized as $MnBi_{2n}Te_{3n+1}$, where n refers to the stoichiometric number of the Bi element. Yan et al. grew $MnBi_2Te_4$ crystal via a flux method, in which the crystal is typically a few millimeters long and often grown in thick, block-like forms with thicknesses up to 2 mm, but which can be easily exfoliated to thin sheets [172]. $MnBi_2Te_4$ has a rhombohedral crystal structure with A-B-C accumulation along the crystallographic *c*-axis of the septuple layers, which can be viewed as inserting one MnTe layer into a quintuple layer. Similarly, the structure of $MnBi_4Te_7$ (n = 2) and $MnBi_6Te_{10}$ (n = 3) also can be regarded as arising from the stacking of quintuple and septuple layers (Figure 7a). Li et al. successfully obtained high-quality $MnBi_2Te_4$ single crystals with a clear and complete antiferromagnetic transition at 25 K from a 1:1 mixture of $Bi_2Te_3$ and MnTe, which were synthesized by a direct reaction. They characterized the magnetic properties of the single crystals of $MnBi_2Te_4$ by zero-field cooling (ZFC) in SQUID [173]. The temperature-dependent magnetic susceptibility of the $MnBi_2Te_4$ single crystals in out-of-plane (*H*//c) and in-plane (*H*//ab) magnetic fields were clearly observed. As the temperature decreased, the magnetic susceptibility of $MnBi_2Te_4$ in *H*//c sharply increased until it reaches a maximum value at 25 K, and then began to decline slowly. This antiferromagnetic transition corresponded to the antiferromagnetic sequence produced by the exchange coupling of $Mn^{2+}$ in the adjacent septuple layers, showing a Néel temperature ($T_N$) of 25 K. In particular, when the temperature was lower than $T_N$, the susceptibility of *H*//c was significantly lower than $\chi_{maximum}$, while the susceptibility of *H*//ab was only slightly lower than $\chi_{maximum}$. Such an obvious difference between in-plane and out-of-plane susceptibility indicates an anisotropy of antiferromagnetic order, which was further revealed by the magnetization field dependencies of *H*//c and *H*//ab (Figure 7b–e).

Yan et al. systematically studied the evolution of structural, magnetic, and transport properties in the whole compositional range of $MnBi_{2-x}Sb_xTe_4$ [174]. Their work suggests that the interlayer exchange coupling, single-ion anisotropy, and magnon gap also decreased with the substitution of Bi by Sb. Different stacking of quintuple and septuple layers enriches the expected topological phenomena in $MnBi_{2n}Te_{3n+1}$ compounds [175]. In theory, $MnBi_4Te_7$ and $MnBi_6Te_{10}$ are considered possible material platforms for chiral Majorana fermions. Yan et al. grew $MnBi_4Te_7$ and $MnBi_6Te_{10}$ out of a Bi-Te flux. The magnetic, transport, and thermodynamic properties measurements of $MnBi_4Te_7$ and $MnBi_6Te_{10}$ single crystals indicated that both $MnBi_4Te_7$ and $MnBi_6Te_{10}$ formed an A-type antiferromagnetic structure with ferromagnetic layers coupled antiferromagnetically [176]. According to the relationship between magnetic susceptibility and temperature measured in an external magnetic field of 250 Oe perpendicular to (labeled *H*//ab) and parallel to (*H*//c) the sample, they drew the conclusion that $MnBi_4Te_7$ and $MnBi_6Te_{10}$ possessed similar

anisotropic temperature-dependent magnetic susceptibility, in which the Néel temperatures ($T_N$) of $MnBi_4Te_7$ and $MnBi_6Te_{10}$ were 13 K and 11 K, respectively (Figure 7f) [176].

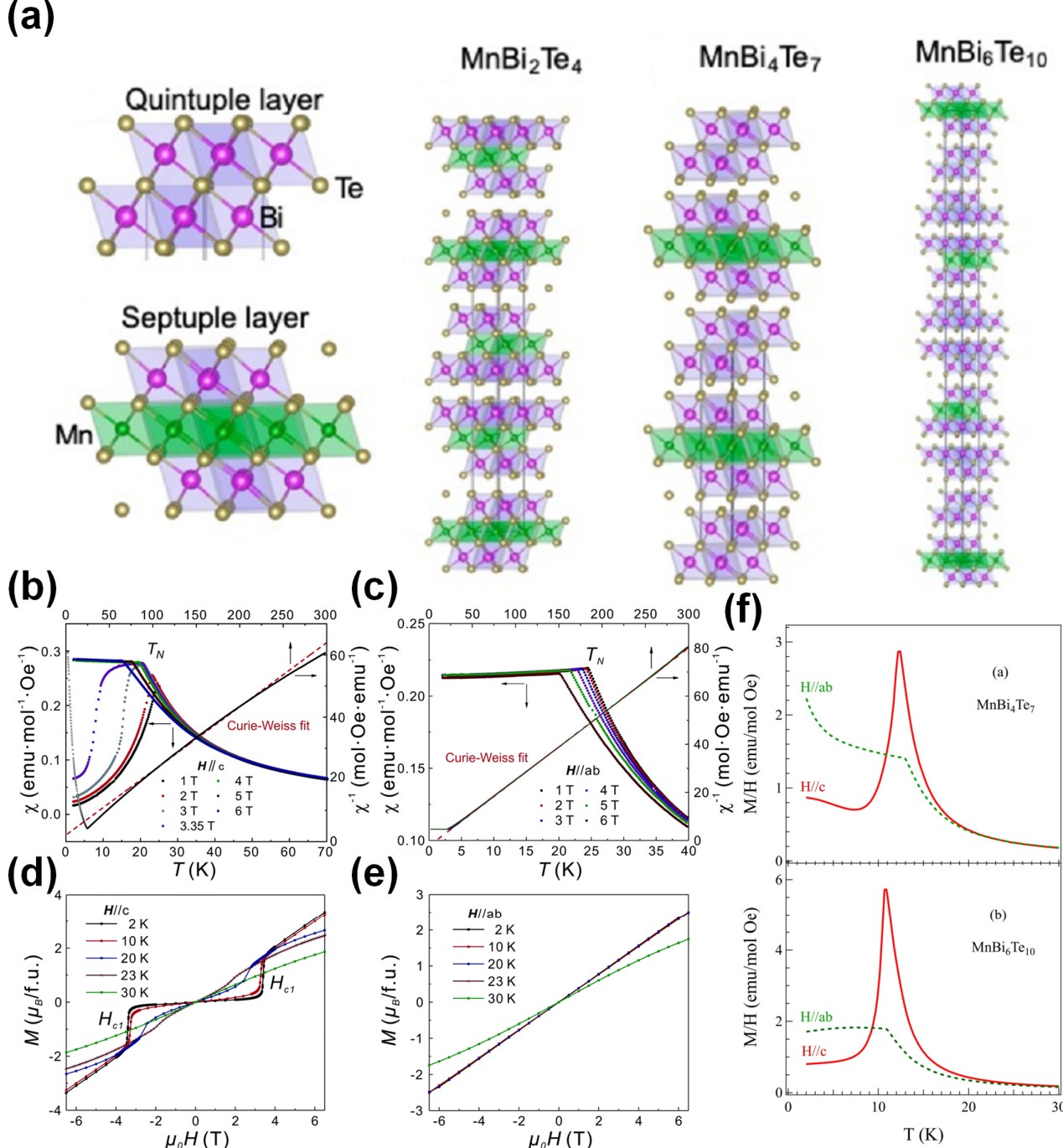

**Figure 7.** (**a**) Structure units of $MnBi_{2n}Te_{3n+1}$. Reused with permission from [176]. (**b,c**) Magnetic properties of single-crystal $MnBi_2Te_4$. Temperature dependence of magnetic susceptibility for the $MnBi_2Te_4$ single crystal and inverse magnetic susceptibility as a function of temperature. The magnetic field is out-of-plane (**a**) and in-plane (**b**), respectively. Reused with permission from [173]. (**d,e**) Field dependence of magnetization for the $MnBi_2Te_4$ crystal measured at different temperatures. The magnetic field is out-of-plane (**d**) and in-plane (**e**), respectively. Reused with permission from [173]. (**f**) Temperature dependence of magnetic susceptibility under an applied magnetic field

of 250 Oe perpendicular (*H//ab*) and parallel (*H//c*) to the crystallographic *c*-axis for $MnBi_4Te_7$ (upper panel) and $MnBi_6Te_{10}$ (lower panel). Reused with permission from [176].

We have systematically summarized the properties of 2D vdW magnetic materials mentioned above with respect to their thicknesses in Table 1.

**Table 1.** Transition temperature, synthesis technique, intrinsic or extrinsic magnetic orderings, electrical properties of 2D vdW magnetic materials.

| Materials | Transition Temperature | Synthesis Technique | Magnetic Orderings | Intrinsic /Extrinsic | Electrical Properties | Characterization Technique |
|---|---|---|---|---|---|---|
| $Cr_2Ge_2Te_6$ [16, 52, 57] | $T_C$ = 61 K (Bulk) $T_C$ = 30 K (Bilayer, under 0.075 T) | Solid State Reaction/Exfoliation | FM | Intrinsic | Insulator | Superconducting Quantum Interference Device (SQUID), Magneto-optical Kerr effect (MOKE) |
| $CrSe_2$ [58] | $T_C$ = 65 K (Monolayer) | CVD | FM | Intrinsic | Semiconductor | Physical Property Measurement System (PPMS, Quantum Design), Raman spectroscopy, Refractive Magnetic Circular Dichroism microscopy (RMCD) |
| $CrTe_2$ [59, 60] | $T_C$ = 310 K (Bulk) $T_C$ = 300 K (Few layers) $T_C$ = 200 K (Monolayer) | CVD MBE | FM | Intrinsic | Semimetal (1T) Semiconductor | Physical Property Measurement System (PPMS, Quantum Design), Vibrating Sample Magnetometer (VSM), Superconducting Quantum Interference Device (SQUID) X-ray Absorption Spectroscopy and Magnetic Circular Dichroism, Angle-Resolved Photoemission Spectroscopy and Scanning Tunneling Microscopy |
| $Fe_3GeTe_2$ [55, 64, 65, 67] | $T_C$ = 230 K (Bulk) $T_C$ = 300 K (Flake, under voltage gate) $T_C$ = 130 K (Monolayer) | Solid State Reaction /CVT/Exfoliation | FM | Intrinsic | Metal | Superconducting Quantum Interference Device SQUID magnetometer, Magnetic Properties Measurement System (MPMS, Quantum Design), Refractive Magnetic Circular Dichroism Microscopy (RMCD), Angle-Dependent Anomalous Hall Measurements |
| $Fe_5GeTe_2$ [70] | $T_C$ = 310 K (Bulk) $T_C$ = 270–300 K (Flake) | CVT/Exfoliation | FM | Intrinsic | Unknown | Physical Property Measurement System (PPMS, Quantum Design), refractive magnetic circular dichroism microscopy (RMCD) |
| $CrI_3$ [15] | $T_C$ = 61 K (Bulk) $T_C$ = 45 K (Monolayer) | CVT/Exfoliation | FM | Intrinsic | Insulator | Magneto-Optical Kerr effect (MOKE), |
| $CrBr_3$ [76, 77] | $T_C$ = 37 K (Bulk) $T_C$ = 34 K (Monolayer) | Exfoliation/MBE | FM | Intrinsic | Insulator | Polarization-Resolved Photoluminescence (PL) Spectroscopy |

| | | | | | | |
|---|---|---|---|---|---|---|
| MnSe$_2$ [80] | $T_C$ = 300 K (Monolayer) | MBE | FM | Intrinsic | Metal | Magnetoresistance measurement, Superconducting Quantum Interference Device (SQUID) |
| VSe$_2$ [83-89] | $T_C$ = 300 K (Flake) | CVT/CVD/EBM /Exfoliation | FM | Intrinsic | Metal | Superconducting Quantum Interference Device (SQUID) |
| MnSn [94] | $T_C$ = 54 K (Monolayer) | MBE | FM | Intrinsic | Semiconductor | Superconducting Quantum Interference Device (SQUID) |
| FeSe$_2$ [95, 96] | $T_C$ = 300 K (Bulk) | Hydrothermal Co-Reduction Route /CVD | FM | Intrinsic | Semiconductor | Vibrating Sample Magnetometer (VSM) |
| FeTe$_2$ [95] | $T_C$ = 300 K (Bulk) | Hydrothermal Co-Reduction Route | FM | Intrinsic | Semiconductor | Vibrating Sample Magnetometer (VSM) |
| FeTe [97, 98] | $T_C$ = 235 K (Multilayer Hexagonal FeTe) $T_N$ = 45 K (Multilayer Tetragonal FeTe) | CVD | FM (Hexagonal) AFM (tetragonal) | Intrinsic | Semiconductor | Vibrating Sample Magnetometer (VSM) Superconducting Quantum Interference Device (SQUID) |
| MnBi$_8$Te$_{13}$ [99] | $T_C$ = 10.5 K (Bulk) | Self-flux | FM | Intrinsic | Insulator | Physical Property Measurement System (PPMS, Quantum Design), Magnetic Properties Measurement System (MPMS, Quantum Design), Single-crystal neutron diffraction |
| $\alpha$/$\beta$-In$_2$Se$_3$ [100-108] | Unknown | Solution method /Exfoliation/PVD/PVT | FM | Extrinsic | Semiconductor | Theoretical calculation |
| MoS$_2$ [109-122] | $T_C$ = 300 K (Few-layer, fluorine adsorption) $T_C$ = 300 K (Bulk, under electron irradiation) | CVD/Exfoliation | FM | Extrinsic | Semiconductor | Theoretical calculation/Superconducting Quantum Interference Device (SQUID) |
| PtSe$_2$ [123-125] | $T_C$ ≈ 5 K (~9 nm thick, Defect induced) | TAC process /Exfoliation | FM | Extrinsic | Semiconductor (Monolayer) Semimetal (Bulk) | Theoretical calculation/Hysteretic Magneto-Transport Response |
| PdSe$_2$ [126-128] | $T_C$ = 800 K (Monolayer, | Solid-State Reaction/Exfoliation | FM | Extrinsic | Semimetal | Theoretical calculation |

| | under uniaxial stress) | | | | | |
|---|---|---|---|---|---|---|
| VTe$_2$ [132-134] | Unknown | Solid-State Reaction/Liquid Exfoliation/MBE | FM | Extrinsic | Semimetal | Theoretical calculation |
| WS$_2$ [135, 136, 140, 141] | $T_C$ = 470 K (Monolayer) | CVD | FM | Extrinsic | Semiconductor | Theoretical calculation |
| SnS$_2$ [142, 143] | $T_C$ = 31 K (Monolayer, Fe-doped) | CVD/CVT/Exfoliation | FM | Extrinsic | Semiconductor | Vibrating Sample Magnetometer (VSM) |
| GaSe [144-149] | Unknown | vdW Epitaxy/CVT | FM (Hole doped) AFM (Fe doped) | Extrinsic | Semiconductor | Theoretical calculation |
| GeSe [150-152] | Unknown | Solid-State Reaction Bridgman method | AFM | Intrinsic | Semiconductor | Theoretical calculation Four-Probe method |
| FeGa$_2$S$_4$ [156, 160] | $T_N$ = 160 K (Bulk) | CVT | AFM | Intrinsic | Insulator | Superconducting Quantum Interference Device (SQUID) |
| NiGa$_2$S$_4$ [159, 160] | $T_N$ = 80 K (Bulk) | CVT | AFM | Intrinsic | Insulator | Superconducting Quantum Interference Device (SQUID) |
| Fe$_2$Ga$_2$S$_5$ [160] | $T_N$ = 130 K (Bulk) | CVT | AFM | Intrinsic | Insulator | Superconducting Quantum Interference Device (SQUID) |
| FePS$_3$ [161, 166] | $T_N$ = 123 K (Bulk) $T_N$ = 118 K (Monolayer) | CVT/Exfoliation | AFM | Intrinsic | Semiconductor | Raman spectroscopy |
| NiPS$_3$ [162, 166] | $T_N$ = 155 K (Bulk) | CVT | AFM | Intrinsic | Semiconductor | Raman spectroscopy/Superconducting Quantum Interference Device (SQUID) |
| MnPS$_3$ [163, 166] | $T_N$ = 78 K (Bulk) | CVT | AFM | Intrinsic | Semiconductor | Vibrating Sample Magnetometer (VSM) |
| MnBi$_2$Te$_4$ [172, 173] | $T_N$ = 25 K (Bulk) | Self-fulx | AFM | Intrinsic | Insulator | Superconducting Quantum Interference Device (SQUID) |
| MnBi$_4$Te$_7$ [176] | $T_N$ = 13 K (Bulk) | Self-fulx | AFM | Intrinsic | Insulator | Magnetic Property Measurement System (MPMS, Quantum Design), Neutron diffraction |
| MnBi$_6$Te$_{10}$ [176] | $T_N$ = 11 K (Bulk) | Self-fulx | AFM | Intrinsic | Insulator | Magnetic Property Measurement System (MPMS, Quantum Design), Neutron diffraction |

## 5. Synthesis Techniques, Properties, Property Manipulation, and Devices Based on Ferromagnetism or Antiferromagnetism

### 5.1. Synthesis Techniques

Large area growth is one of the main challenges of 2D materials, and impedes their practical application. To date, mechanical exfoliation is still one of the most widely

adopted methods for obtaining 2D materials. Through mechanical exfoliation, researchers obtained 2D graphene for the first time [177]. Generally, mechanical exfoliation can obtain high-quality 2D materials (Figure 8a) [178]. However, it is inefficient and cannot be integrated with current industrial practices. More importantly, mechanical exfoliation is not suitable for all kinds of 2D materials. Recently, Huang et al. proposed a new strategy, in which mechanical exfoliation was assisted by an Au adhesion layer with covalent-like quasi-bonding to produce a layered crystal; this method provides access to a broad spectrum of large-area monolayer materials. Significantly, such an approach is applicable to a large number of 2D materials (Figure 8b) [179]. Similarly, Moon et al. developed a layer-engineered exfoliation (LEE) approach to obtain extraordinarily large-sized and high-density graphene from natural graphite. In this method, a selective metal film was directly evaporated on bulk graphite precleaved on blue tape and then large-area graphene with a selected number of layers can be exfoliated easily. This LEE approach paves the way for the development of a manufacturing-scale process for future applications based on a 2D heterostructure (Figure 8c) [180].

Liquid exfoliation is another general method for synthesizing high-quality 2D materials. Coleman et al. demonstrated that several-layer materials, such as $MoS_2$, $MoSe_2$, $WS_2$, and BN, can be efficiently dispersed in common solvents and deposited as individual flakes, or can be formed into films (Figure 8d) [181]. In this method, the choice of solvent is the key factor. Zeng et al. created an electrochemical way to fabricate few-layer-thick 2D materials such as BN, $NbSe_2$, and $WSe_2$. In their method, lithium intercalation of the layered material was performed in a test cell with Li foil as an anode and 1 M $LiPF_6$ dissolved in a mixture of ethyl carbonate (EC) and dimethyl carbonate (DMC) (1:1 in volume ratio) as an electrolyte. The layered bulk material was prepared as a cathode by adding an acetylene black and poly (vinylidene fluoride) (PVDF) binder dispersed in N-methylpyrrolidone (NMP) solutions. After the discharge process, the lithium-intercalated sample was washed with acetone to remove the residual electrolyte ($LiPF_6$), followed by exfoliation and sonication in $N_2$-saturated Milli-Q water in a closed vial (Figure 8e) [182].

CVD is also one of the most widely adopted methods for synthesizing 2D materials, and is considered one of the most suitable methods for large-area growth. In 2006, Somani et al. experimentally demonstrated for the first time that planar, few-layer nanographenes (PFLGs) could be synthesized by a simple thermal CVD method [183]. In recent years, the growth of 2D materials based on the CVD method has been developed extensively. Via the CVD method, many kinds of high-quality 2D crystals, continuous films, and even heterostructures have been grown with large-scale production capability and good controllability, which are of great significance for bringing practical applications to fruition (Figure 8f) [184]. MBE is another efficient way to synthesize ultrathin films, and exhibits great superiority with respect to controlling the thickness of the materials. In previous reports, some 2D materials, such as $CrBr_3$ (Figure 8g) [77], $CrTe_2$ [60], $MnSe_2$ [80], and MnSn [94], were successfully synthesized via MBE. However, the costly facilities and complex processes hinder its development in the growth of 2D materials.

### 5.2. Quantum Anomalous Hall Effect (QAHE)

The quantum anomalous Hall effect (QAHE) has been experimentally observed in previous studies, and is a unique transport phenomenon of the magnetic topological insulator that has become an important research field in condensed matter physics [185]. In a quantum anomalous Hall insulator, the spontaneous magnetic moment combined with spin–orbit coupling produces a topologically nontrivial band structure, which shows QAHE in the absence of a magnetic field [186]. In 2013, Chang et al. reported the observation of the QAHE in thin films of chromium-doped $(Bi, Sb)_2Te_3$, a magnetic topological insulator [22]. Zhang et al. systematically investigated the QAHE of graphene deposited on the (001) surface of an antiferromagnetic insulator $RbMnCl_3$ from first-principles calculation [23]. Recently, research on QAHE materials has mainly focused on intrinsic magnetic topological insulators (MTIs), including $MnBi_2Te_4$ and related heterostructures

constructed of $MnBi_2Te_4$ and $Bi_2Te_3$ [187]. 2D $MnBi_2Te_4$ has an in-plane ferromagnetic order and out-of-plane antiferromagnetic order, the magnetic properties and topological phases of which change as a function of the number of layers [188]. Additionally, Li et al. found that $MnBi_2Te_4$ with even layers is an axion insulator due to the loss of net magnetization, while $MnBi_2Te_4$ with odd layers is a candidate for an ideal Chern insulator to realize QAHE [189].

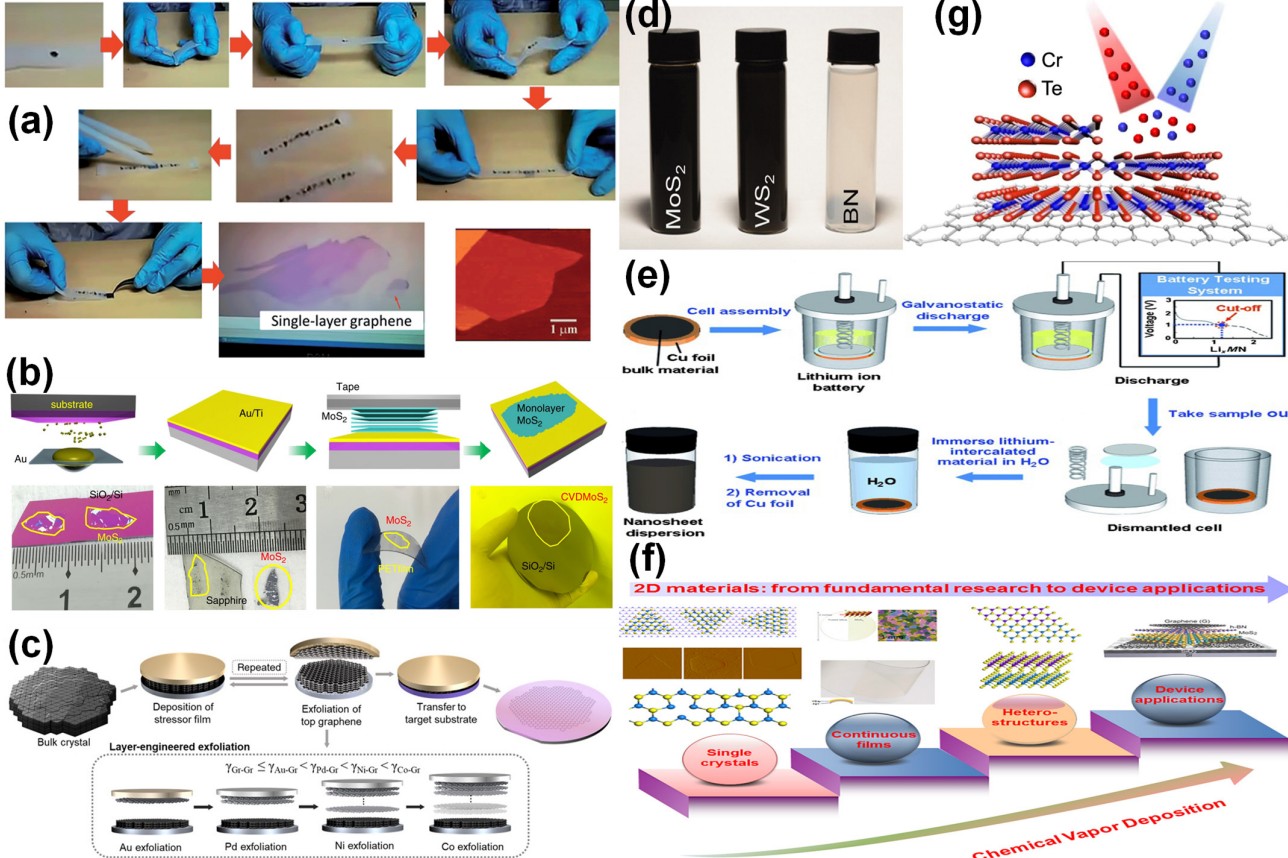

**Figure 8.** (**a**) Illustrative procedure of the micromechanical cleavage of graphene based on transparent tape. Reused with permission from [178]. (**b**) Schematic of the exfoliation process assisted by an Au adhesive. Reused with permission from [179]. (**c**) Schematic illustration of the LEE large-area graphene exfoliation technique. Reused with permission from [180]. (**d**) Photographs of the dispersions of $MoS_2$ (in N-methyl-pyrrolidone (NMP)), $WS_2$ (in N-methyl-pyrrolidone (NMP)), and BN (in isopropanol (IPA)). Reused with permission from [181]. (**e**) The electrochemical lithium intercalation process is used to produce 2D nanosheets from the layered bulk material. Reused with permission from [182]. (**f**) Overview of the CVD growth of 2D materials, from single crystals to continuous films, and further to 2D-material-based heterostructures, as well as their device applications. Reused with permission from [184]. (**g**) Illustration of MBE growth process of $CrTe_2$ films on graphene. Reused with permission from [60].

### 5.3. Magnon

Magneton or magneto-based spintronics, which primarily include the dynamic behavior of spin waves in nanomaterials, is also a transport phenomenon in 2D magnetism [163,190]. To explore magnon-mediated transport in 2D antiferromagnetic $MnPS_3$, a device was fabricated, as demonstrated in Figure 9a, where the left Pt electrode acted as a magnon injector and the right Pt electrode acted as a magnon detector [163]. The magnon transport and relaxation characteristics of $MnPS_3$ were then systematically measured at 2,

5, and 10 K, from which long-distance magnon transport in MnPS₃ was observed (Figure 9b). It can be concluded that the larger the diffusion effect of the magnons on the crystal's surface, the greater the diffusion effect of impurities on the crystal's surface. With a decrease in temperature, the diffusion length of the magnons became longer, indicating that the lifetime of a magnon at low temperature was longer. Ghazaryan et al. studied the tunneling effect in graphene/CrBr₃/graphene vdW heterostructures and found that inelastic tunneling through thin ferromagnetic CrBr₃ barrier was mainly related to magnon emission at low temperature; that is, the final state in which the spin state (up or down) of the tunneling electron will flip between the initial state and the final state during the magnon tunnelling process [190]. Tunneling effects with magneton emission and proximity effect provide a new prospect for the development of 2D ferromagnetic barriers in graphene spintronics.

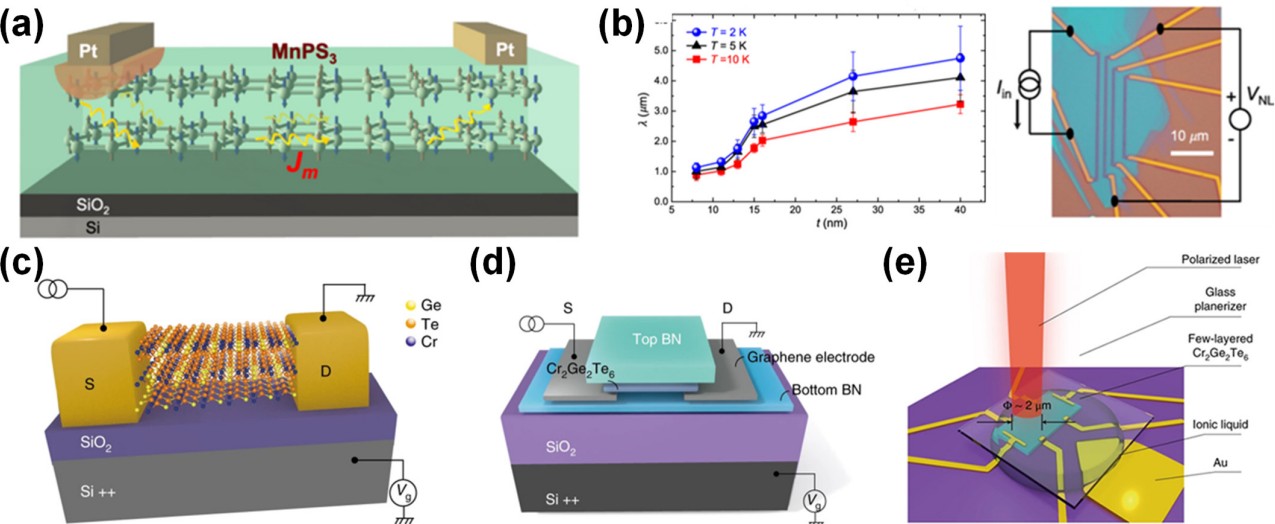

**Figure 9.** (**a**) Scheme of nonlocal measurements made of an MnPS₃ device. Reused with permission from [163]. (**b**) Scheme of long-distance magnon transport of MnPS₃ (right panel), and magnon relaxation distance as a function of crystal thickness in quasi-2D MnPS₃ at 2, 5, and 10 K (left panel). Reused with permission from [163]. Schematics of (**c**) an Au-contacted, few-layered Cr₂Ge₂Te₆ device and (**d**) a few-layered Cr₂Ge₂Te₆ flake encapsulated by two h-BN layers, contacted via graphene electrodes. (**e**) Experimental setup for Kerr measurement of the micrometer-sized device using an ionic liquid. Reused with permission from [61].

### 5.4. Magnetic Skyrmions

Magnetic skyrmions, sometimes also called vortices [191] or vortex-like [192], were first predicted in theory [191,193,194], and have been characterized experimentally by means of neutron scattering in momentum space and microscopy techniques in real space in recent years [195–197]. Magnetic skyrmions are nanometer-sized quasiparticles [198] defined as a local whirl of the spin configuration in a magnetic material. Compared with a single magnetic domain, magnetic skyrmions possessed high energetic stability (per unit volume) and allow the existence of discrete magnetic states. Therefore, magnetic skyrmions acting as bits to store information show promise to be used in future high-performance memory [199–201] and logic [202,203]. Additionally, dynamical magnetic skyrmions exhibit an impressive robust breathing feature, which means skyrmion-based microwave application is an achievable reality [204]. However, there are some intriguing and important questions that must be investigated clearly, such as its creation and annihilation processes, how to control the size of magnetic skyrmions, more accurate understanding of the role of defects in magnetic skyrmions, etc.

### 5.5. Property Manipulation

Currently, chemical doping, defect engineering, and the magnetic proximity effect have been adopted to obtain higher $T_C$ values, and certain progress has been made. Yang et al. experimentally demonstrated a novel, two-step tellurium flux strategy to introduce magnetic Cr atoms into layered semimetal $T_d$-$WTe_2$ to realize highly tunable, near-room temperature ferromagnetism, in which the semimetal properties of the Cr-doped layered $T_d$–$WTe_2$ were intriguingly preserved [205]. Li et al. successfully synthesized Mn-doped $CdSe(en)_{0.5}$ via a solvothermal process at 140 °C, in which they observed giant Zeeman splitting and zero-field splitting effect at room temperature [206]. Their work suggested that 2D monolayer nanosheets like Mn-doped $CdSe(en)_{0.5}$ had advantages for future diluted magnetic semiconductor (DMS) applications. Recently, Ge et al. reported magnetic moments induced by atomic vacancies in transition metal dichalcogenide $PtSe_2$ flakes, and they experimentally demonstrated the presence of nearly thickness-dependent, localized magnetic moments induced by Pt-vacancy defects in air-stable $PtSe_2$ flakes [207]. Their work indicated that defect control was a practical approach to induce magnetism in non-magnetic materials, offering a new way to modulate magnetism in TMDCs.

However, such research is still far away from practical applications. The emergent ferromagnetism near three-quarters filling of the conduction miniband in twisted bilayer graphene [208,209], as well as a tunable correlated Chern insulator and ferromagnetism in ABC-trilayer graphene/hexagonal boron nitride (ABC-TLG/hBN) Moiré superlattice [210], have experimentally verified that layer engineering may be an effective method to achieve the ambient-temperature application of 2D ferromagnetism in the future. The experimental research of Zhang et al. indicated that proximity coupling will induce a significant enhancement of the coercive field and Curie temperature in 2D vdW heterostructure [211]. This work revealed that the proximity coupling of $FePS_3$ with $Fe_3GeTe_2$ is an effective method for enhancing the ferromagnetism of 2D GeFeTe, which paved the way for applications in advanced magnetic spin electronic devices and memory devices. In addition, Wang et al. observed a significant enhancement of the coercive field in vdW magnet $Fe_3GeTe_2$ (FGT) under strain [212]. Moreover, Li et al. also experimentally achieved strong ferromagnetism via breathing lattices in atomically thin cobaltites, which indicated that strain engineering and interface engineering were potentially feasible ways to improve the properties of 2D ferromagnetic materials [213]. Additionally, Zhang et al. experimentally realized a large current control of the coercive field and magnetic memory based on nanometer-thin ferromagnetic vdW $Fe_3GeTe_2$ (FGT), potentially opening up a fascinating avenue of electrical modulation and spintronic applications using 2D magnetic vdW materials [214]. Notably, materials with strong magnetoresistance responses are the basis of spintronic technology, magnetic sensors, and hard disk drives. In earlier studies, manganese oxides with mixed-valence and cubic perovskite structures stood out due to their CMR, which was attributed to the existence of double exchange interaction in manganates with the enhanced charge transport when the spins on neighboring $Mn^{3+}$ and $Mn^{4+}$ ions were parallel [215–218]. Recently, Wang et al. experimentally discovered CMR without mixed valence in a layered phosphide $EuCd_2P_2$ crystal, in which the absence of direct Cd-Cd bonds in the structure of $EuCd_2P_2$ seemed to be beneficial to CMR [219]. In terms of stability, based on Ising transition theory, the stability of long-range, robust magnetic order could be enhanced by introducing anisotropy.

### 5.6. 2D vdW Heterostructures and FET

Owing to their dangling bond-free property, 2D materials are considered an ideal platform for constructing heterostructures, offering great opportunities to realize many interesting phenomena [220]. One great advantage of 2D magnetic heterostructures is that the proximity-induced exchange interaction can fundamentally change the electronic structure of the 2D system. There is no need to consider lattice mismatch for vdW heterostructures; thus, they can almost keep their original layered structure. By changing the

interlaminar twist angle or stacking order, there are unlimited possibilities for fabricating multifarious vdW heterostructures consisting of diverse 2D ferromagnets, which offer new ways to study and design magnetic properties. A $MoS_2$-$WS_2$ bilayer is a typical example of bandgap engineering in atomic, thin 2D semiconductor heterostructures [221]. The work of Hou et al. proved that the topological surface state of $Bi_2Se_3$ could be maintained and magnetized by forming $CrI_3$/$Bi_2Se_3$/$CrI_3$ heterostructures [222]. Jiang et al. successfully prepared double-gate graphene/BN/$CrI_3$ field-effect devices, the magnetic properties of which were effectively controlled by adjusting the gate voltage [74]. Generally, there are three typical device structures of 2D-material-based FETs, in which $SiO_2$, ionic liquids, and h-BN are used as dielectric layers. Figure 9c–e illustrate three FET configurations based on a few-layer ferromagnetic semiconductor, $Cr_2Ge_2Te_6$ [61].

### 5.7. Spintronic Applications

Spintronics-based devices exhibit higher efficiency of data storage, transmission, and calculation. Significantly, the discovery of 2D vdW magnets is beneficial for fabricating various spintronics devices, including spin valves, spin–orbit torque (SOT), spin field-effect transistors, and spin filters. Based on the conducting ferromagnet $Fe_3GeTe_2$ [223], a simple $Fe_3GeTe_2$/h-BN/$Fe_3GeTe_2$ heterostructure spin valve device was experimentally fabricated, in which if the magnetization directions of the two electrodes were opposite (parallel), the tunneling resistance showed typical spin valve behavior. SOT is a new method for controlling the magnetization of ferromagnetic materials by injecting an in-plane current through large spin-orbit coupling [224]. A serious ferromagnetic motion will occur in a system composed of gravity and magnetic layers. In the traditional SOT bilayer system, the magnetized switch depends on the intrinsic spin Hall effect and the interface mass [225]. Shao et al. fabricated a SOT device constructed of a transition metal dihydroxy compound monolayer ($MX_2$: M = Mo, S; X = S, Se) and an interfacial ferromagnetic CoFeB, which graphically illustrated spin accumulation at the $MX_2$/CoFeB interface due to the Rashba–Edelstein effect under an applied electric field [226].

Transistor devices based on spin rather than charge can realize high performance in nonvolatile memory applications, in which ferromagnetic materials act as a drain/source electrode for spin polarizer and an electronic analyzer along the current direction. Longo et al. successfully synthesized Fe/Au/$Sb_2Te_3$ heterostructures via metal organic chemical vapor deposition (MOCVD). Through broadband ferromagnetic resonance spectroscopy (BFMR) and spin pumping FMR (SP-FMR), they experimentally observed the spin behavior of large-area topological insulators exhibiting spin to charge (S2C) with the inverse Edelstein effect (IEE). They found a conversion efficiency of $\lambda_{IEE} \approx 0.27$ nm for Fe/Au/$Sb_2Te_3$, originating from the topologically protected surface states (TSS) of $Sb_2Te_3$ [227]. Recently, a spin field-effect transistor (SFET) composed of a 2D 2H-$VSe_2$ magnet was proposed, which could exert a vertical electric field to convert an A-type antiferromagnet into a semimetal [228].

## 6. Summary and Outlook

In summary, we have reviewed the latest research progress on 2D magnets, including material systems with intrinsic and induced ferromagnetism or antiferromagnetism, as well as their physical properties, device fabrication, and potential applications. The 2D ferromagnets are considered ideal candidates for next-generation spintronic devices, offering a promising platform for novel quantum phenomenon such as QAHE. However, weak ferromagnetic coupling and low Curie temperature ($T_C$) are the key obstacles that need to be overcome in the field of 2D ferromagnets. Some methods, such as chemical doping, defect engineering, and magnetic proximity effect, to some extent, have been demonstrated to be effective ways to improve magnetic properties; however, there are still many challenges not only in theory but also in technology for developing room-temperature 2D ferromagnets with strong ferromagnetic coupling. Layer engineering, strain

engineering, and interface engineering are considered to be promising ways to improve the properties of 2D ferromagnetic materials in the future.

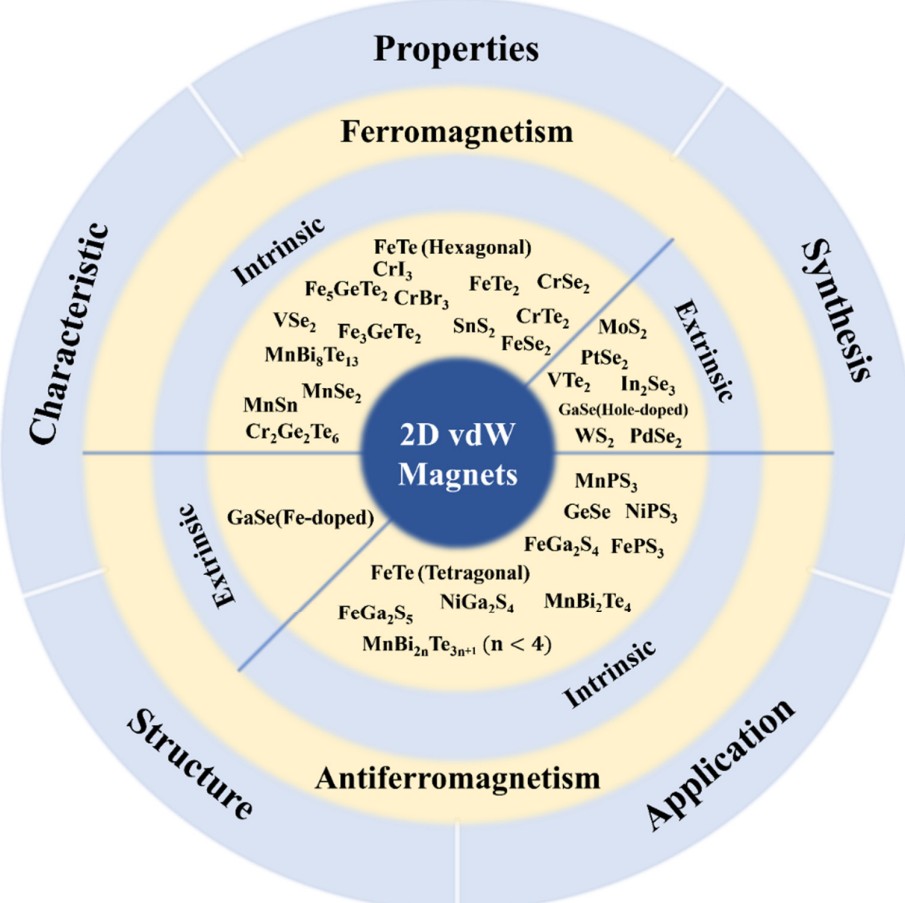

**TOC**: Based on the magnetic properties of 2D vdW materials, this review summarizes most known materials with intrinsic/extrinsic antiferromagnetism or ferromagnetism, including their structures, characteristics, properties, fabrication methods, and applications.

**Author Contributions:** W.H. and L.K. have contributed equally to this work. W.H. and L.K. wrote this review under the supervision of P.Y. and W.Z. All authors have read and agreed to the published version of the manuscript.

**Funding:** This work was funded by the National Key Research and Development Program of China (No. 2021YFE0194200 and No. 2021YFA1200903), the National Natural Science Foundation of China (No. 22175203 and No. 22006023), Natural Science Foundation of Guangdong Province (No. 2019A1515010428 and No. 2020A1515110821), and Guangzhou Science and technology project (No. 202102020126). This work was also funded by the Plan Fostering Project of State Key Laboratory of Optoelectronic Materials and Technologies, of Sun Yat-sen University (No. OEMT-2021-PZ-02).

**Acknowledgments:** This work was supported by the National Key Research and Development Program of China (No. 2021YFE0194200 and No. 2021YFA1200903), the National Natural Science Foundation of China (No. 22175203 and No. 22006023), Natural Science Foundation of Guangdong Province (No. 2019A1515010428 and No. 2020A1515110821), and Guangzhou Science and technology project (No. 202102020126). This work was also supported by the Plan Fostering Project of State Key Laboratory of Optoelectronic Materials and Technologies, of Sun Yat-sen University (No. OEMT-2021-PZ-02).

**Conflicts of Interest:** The authors declare no conflict of interest.

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
