# Peer review of "Atomically Thin 2D van der Waals Magnetic Materials: Fabrications, Structure, Magnetic Properties and Applications"

_coatings, doi:10.3390/coatings12020122_

Round 1

Reviewer 1 Report

The present review article is written on 2D van der Waals Magnetic materials. It is a very well-written paper and the way of presentation is also good. Maybe authors can incorporate the following comments.

  1. Please add the chalcogenides materials, which is Fe, Ge, and Sn series (Gr III and Gr IV materials in detail)
  2. Kindly improve the figure resolution if Figure 3 (g), Figure 4(d) (upper panel).
  3. In the summary, please redraw the pi-diagram in different way (if possible). The message from the figure is not clear.
  4. Maybe it will be very informative if authors add a table demonstrating the material magnetic, and electronic properties with respect to their thickness.
  5. Kindly add the following references:
  7. Duong et al. ACS Nano 2017, 11, 11803−11830

Author Response

Reviewer #1The present review article is written on 2D van der Waals Magnetic materials. It is a very well-written paper and the way of presentation is also good. Maybe authors can incorporate the following comments.

  • Please add the chalcogenides materials, which is Fe, Ge, and Sn series (Gr III and Gr IV materials in detail)

Reply: We thank the reviewers for your kind advice. We have added add the chalcogenides materials, which is Fe, Ge, and Sn series (Gr III and Gr IV materials in detail) in main text.

3.1.5 MnSn

In early theoretical study, bulk MnSn in zinc-blende structure has been predicted as a half-metallic ferromagnet, which was anticipated to be valuable in spintronics owing to its lattice compatibility with semiconductors [85, 86]. Yuan et al. successfully synthesized ultrathin MnSn film from monolayer to few layers by molecular beam epitaxy (MBE), in which the clear magnetic behavior was observed experimentally [87]. Through superconducting quantum interference device (SQUID) magnetometer (Quantum Design), configured with a longitudinal pick-up coil, the magnetic behavior shows clear thickness dependence, which provides a strategy to precisely tune the magnetism and a new platform to study 2D magnetism, and the possibility to integrate 2D ferromagnetism with silicon technology.

In another work, Li et al. synthesized Fe-doped SnS2 bulk crystals via direct vapor-phase method and subsequently obtained monolayer Fe-doped SnS2 through mechanical exfoliation. They found that monolayer Fe-doped SnS2 exhibits ferromagnetic behavior a perpendicular anisotropy at 2 K and a Curie temperature of ∼31 K.

3.1.6 FeB2 (B = Se, Te)

In previous study, Liu et al. developed a hydrothermal co-reduction method to fabricate ferroselite (FeSe2) nanorods and frohbergite (FeTe2) nanocrystallites using N2H4·H2O as the reductant. They observed the distinct magnetic hysteresis loop of FeSe2 and FeSe2 via vibrating sample magnetometer (VSM) [88]. Very recently, the sublimed-salt-assisted CVD method at atmospheric pressure (APCVD) was adopted to synthesize the layered FeSe2 nanocrystals, which exhibit a phase transition at ≈11 K [89]. Recently, Kang et al. realized Phase-controllable growth of ultrathin 2D magnetic FeTe crystals. According to their work, antiferromagnetic tetragonal FeTe with the Néel temperature (TN) gradually decreases from 70 to 45 K as the thickness declines from 32 to 5 nm, and ferromagnetic hexagonal FeTe is accompanied by a drop of the Curie temperature (TC) from 220 K (30 nm) to 170 K (4 nm) [90]. Meanwhile, through chemical vapor deposition, Wang et al. successfully synthesized ultrathin FeTe nanosheets with tetragonal and hexagonal phases by tuning the growth temperature according to the phase diagram. The magneto-transport measurements illustrate that the tetragonal crystals display a linear magnetoresistance (LMR) as high as 10.5% at 1.9 K and hexagonal nanosheets show 5.8% LMR at 1.9 K, indicating their potential applications as magnetoresistive devices. Moreover, tetragonal FeTe illustrates a superconducting transition at 9.0 K, may be caused by oxygen doping, indicating that FeTe is an intriguing material platfrom to explore 2D magnetism, superconductivity, and their heterostructures for potential applications in novel, functional devices. [91].”

  1. Yu, L.; Yao, K.; Liu, Z.; et al. Electronic structure and magnetic property of MnSn: Prediction of half-metallic ferromagnetism in zinc-blende structure. Solid State Commun. 2007, 144, 18-22.
  2. Li, J.; Li, Y.; Dai, X.; et al. Ab initio investigation of half-metal state in zinc-blende MnSn and MnC. Physica B Condens. Matter 2008, 403, 2473-2476.
  3. Yuan, Q.-Q.; Guo, Z.; Shi, Z.-Q.; et al. Ferromagnetic MnSn monolayer epitaxially grown on silicon substrate. Chinese Phys. Lett. 2020, 37, 077502.
  4. Liu, A.; Chen, X.; Zhang, Z.; et al. Selective synthesis and magnetic properties of FeSe2 and FeTe2 nanocrystallites obtained through a hydrothermal co-reduction route. Solid State Commun. 2006, 138, 538-541.
  5. Liu, H.; Xue, Y. Van Der Waals Epitaxial Growth and Phase Transition of Layered FeSe2 Nanocrystals. Adv. Mater. 2021, 33, 2008456.
  6. Kang, L.; Ye, C.; Zhao, X.; et al. Phase-controllable growth of ultrathin 2D magnetic FeTe crystals. Nat. Commun. 2020, 11, 1-9.
  7. Wang, X.; Bian, C.; He, Y.; et al. Ultrathin FeTe nanosheets with tetragonal and hexagonal phases synthesized by chemical vapor deposition. Mater. Today 2021, 45, 35-43.

3.2.7 SnS2

In previous study, Su et al. proposed a novel strategy for controlled CVD synthesis of thin crystal arrays of layered semiconductors SnS2, in which the few-layer SnS2 was demonstrated as ultrasensitive photodetectors [135]. In another work, Li et al. synthesized Fe-doped SnS2 bulk crystals via direct vapor-phase method and subsequently obtained monolayer Fe-doped SnS2 through mechanical exfoliation. Via a vibrating sample magnetometer (VSM) from a physical properties measurement system (PPMS), they found that monolayer Fe-doped SnS2 exhibits ferromagnetic behavior a perpendicular anisotropy at 2 K and a Curie temperature of ∼31 K [136].

3.2.8 GaSe and GeSe

In early study, Zhou et al. successfully realized controllable synthesis of single-crystalline ultrathin GaSe nanoplates on a flexible mica substrate via van der Waals epitaxy. Individual 2D GaSe nanoplates were found to show typical p-type conductance and good light response characteristics [137]. Concurrently, Li et al. synthesized ultrathin 2D GaSe via chemical vapor transport (CVT) in a tube furnace system, which shows a high photoresponse and FET characteristics [138]. The magnetic behavior of GaSe was reported many times in theory [139-142]. The 2D unit cell of a GaSe monolayer is composed of two Ga atoms and two Se atoms. The two neighboring Ga atoms are vertically stacked, and are sandwiched between two planes of Se atoms. Gao et al. performed first-principles calculations using density functional theory (DFT) in the local density approximation (LDA) for the exchange-correlation function as implemented in the QUANTUM ESPRESSO package. Their theoretical work revealed that the tunable ferromagnetism and half-metallicity in monolayer GaSe can be induced by hole doping [139]. Soon afterwards, Lu et al. systematically investigated the structural and electronic properties of Fe adsorbed monolayer GaSe by using the first-principles DFT calculations. Attributed to the transfer of Fe-3d electrons, the magnetic moments of −0.004 μB and −0.178 μB are induced for the nearest neighbor Ga and Se atoms, respectively, showing an AFM coupling with the adsorbed Fe atom [140]. Similarly, Tang et al. systematically studied the modulation of the electric field on the structural, electronic and magnetic properties of vacancy doped GaSe monolayer by using the first-principles DFT calculations. According to their work, an unpolarized electronic structure in V–Se doped GaSe monolayer, whereas a half-metallicity with a 1 μB magnetic moment in V–Ga doped system. The stability of the magnetic moment is enhanced with an increased vertical electric field along −z direction but is weakened with that along +z direction, which leads to a prompt switch from the magnetic to nonmagnetic state under a +0.6 V/Å electric field [141]. Recently, in another computational study, Ke et al. predicted that the magnetic properties of Fe-doped monolayer GaSe can be controlled by electrical field [142]. In theory, they proved that the spin polarization, magnetic moment, and MAE are strongly dependent on the applied vertical electric field owing to a magnetic configuration modulation in the system.

Bulk GeSe single crystal can be prepared via solid state reaction in an evacuated silica tube [143]. Xue et al. successfully synthesized micrometer-sized GeSe Nanosheets and investigated its’ anisotropic photoresponse properties [144]. Recently, Sarkar et al. synthesized rhombohedral (GeSe)0.9(AgBiSe2)0.1 crystals grown by Bridgman method, which shows ferroelectric instability induced ultralow thermal conductivity and high p-type thermoelectric performance [145]. Early computational research revealed that monolayer GeSe is a semiconductor with direct band gap of 1.16 eV with strong stability, and the electronic band structure and small effective masses of charge carriers are sensitive to the strain [146]. In 2017, Zhao et al. experimentally obtained monolayer GeSe on SiO2/Si substrates by controlled laser thinning method in high vacuum and systematically study the band structure and photoelectric of GeSe Monolayers [147]. However, the investigation of magnetic properties of GeSe is very lacking. Ishihara et al. reported the negative magnetoresistance in GeSe, which they attributed to the carrier scattering by localized magnetic moments [148].”

  1. Su, G.; Hadjiev, V.G.; Loya, P.E.; et al. Chemical vapor deposition of thin crystals of layered semiconductor SnS2 for fast photodetection application. Nano lett. 2015, 15, 506-513.
  2. Li, B.; Xing, T.; Zhong, M.; et al. A two-dimensional Fe-doped SnS2 magnetic semiconductor. Nat. Commun. 2017, 8, 1-7.
  3. Zhou, Y.; Nie, Y.; Liu, Y.; et al. Epitaxy and photoresponse of two-dimensional GaSe crystals on flexible transparent mica sheets. ACS nano 2014, 8, 1485-1490.
  4. Li, X.; Lin, M.-W.; Puretzky, A.A.; et al. Controlled Vapor Phase Growth of Single Crystalline, Two-Dimensional GaSe Crystals with High Photoresponse. Sci. Rep. 2014, 4, 5497.
  5. Cao, T.; Li, Z.; Louie, S.G. Tunable magnetism and half-metallicity in hole-doped monolayer GaSe. Phys. Rev. Lett. 2015, 114, 236602.
  6. Lu, Y.; Ke, C.; Fu, M.; et al. Magnetic modification of GaSe monolayer by absorption of single Fe atom. RSC Adv. 2017, 7, 4285-4290.
  7. Tang, W.; Ke, C.; Fu, M.; et al. Electrically tunable magnetic configuration on vacancy-doped GaSe monolayer. Phys. Lett. A 2018, 382, 667-672.
  8. Ke, C.; Wu, Y.; Guo, G.-Y.; et al. Electrically controllable magnetic properties of Fe-doped GaSe monolayer. J. Phys. D: Appl. Phys. 2019, 52, 175001.
  9. Okazaki, A. The crystal structure of germanium selenide GeSe. J. Phys. Soc. Jpn. 1958, 13, 1151-1155.
  10. Xue, D.J.; Tan, J.; Hu, J.S.; et al. Anisotropic photoresponse properties of single micrometer‐sized GeSe nanosheet. Adv. Mater. 2012, 24, 4528-4533.
  11. Sarkar, D.; Ghosh, T.; Roychowdhury, S.; et al. Ferroelectric instability induced ultralow thermal conductivity and high thermoelectric performance in rhombohedral p-type GeSe crystal. J. Am. Chem. Soc. 2020, 142, 12237-12244.
  12. Hu, Y.; Zhang, S.; Sun, S.; et al. GeSe monolayer semiconductor with tunable direct band gap and small carrier effective mass. Appl. Phys. Lett. 2015, 107, 122107.
  13. Zhao, H.; Mao, Y.; Mao, X.; et al. Band structure and photoelectric characterization of GeSe monolayers. Adv. Funct. Mater. 2018, 28, 1704855.
  14. Ishihara, Y.; Nakada, I. Negative Magnetoresistance of GeSe in a Weak Magnetic Field. Phys. Status Solidi B Basic Res. 1981, 106, K27-K31.

  • Kindly improve the figure resolution if Figure 3 (g), Figure 4(d) (upper panel).

Reply: We are grateful to this constructive suggestion. We have improved the figure resolution if Figure 3 (g), Figure 4(d) (upper panel).

Figure 3. a) Top view of monolayer CrI3, where Cr atoms (red balls) form a honeycomb structure in an edge-sharing octahedral coordination by six I atoms (blue balls), and side view of bilayer CrI3 of the rhombohedral stacking order. Reused with permission from [67] b) MCD versus magnetic field (left) at three representative doping levels at 4 K (top panel) and 50 K (bottom panel); MCD versus magnetic field at 4 K at representative gate voltages (right). Reused with permission from [68] c) The hysteresis loop for bilayer CrBr3. Reused with permission from [69] d) Crystal structure diagrams of α-MnSe (111) and monolayer 1T-MnSe2. Reused with permission from [73] e & f) Schematic illustration of the electrochemical setup for cathode exfoliation of VSe2 using tetra propylammonium cation (TPA) as the intercalant. Inset: photograph showing a small piece of VSe2 crystal clamped by two titanium plates (left), visible expansion of VSe2 cathode with a fluffy shape (right); M–H hysteresis loop of bare and passivated VSe2 flakes at 300 K. Reused with permission from [84] g) Structure determination and the stacking rule in MnBi2nTe3n+1. Reused with permission from [92]

Figure 4. a) Crystal structure of the Fe–In2Se3 monolayer. Red, yellow, and blue balls represent In, Se, and Fe atoms, respectively. Reused with permission from [96] b) The spin-resolved charge density isosurface (isosurface value at 0.002 e/Å3) of TM-doped MoS2 monolayer at 8 % impurity concentration. Reused with permission from [114] c) Crystal structure of monolayer PtSe2 (left panel, top: unit cell, bottom: top view); band dispersions along the K-G-M-K direction from first-principles calculations. The colour and line width distinguish the contribution from Pt and Se (right panel, Red for Se and green for Pt). Reused with permission from [116] d) Optical and AFM antiferromagnetic images of a completed PtSe2 device and its crystal (upper panel). Magnetic field dependence of ΔR measured from device E (shown in upper panel) at T = 1.5 K and VSD = 10 mV. The red (black) arrow represents the sweep direction from positive (negative) to negative (positive) values. Reused with permission from [7]

  • In the summary, please redraw the pi-diagram in different way (if possible). The message from the figure is not clear.

Reply: We thank the reviewers for your gentle suggestion. We redrew the pi-diagram in a different way.

  • Maybe it will be very informative if authors add a table demonstrating the material magnetic, and electronic properties with respect to their thickness.

Reply: We are grateful to this constructive suggestion. We added a table demonstrating the material magnetic, and electronic properties with respect to their thickness.

“We systematically summarized the recent study of 2D vdW magnetic materials respect to their thickness in Table1.”

Table 1

Transition temperature, synthesis technique, intrinsic or extrinsic magnetic orderings, electrical properties of 2D van der Waals magnetic materials.

Materials

Transition temperature

Synthesis technique

Magnetic orderings

Intrinsic/Extrinsic

Electrical properties

Characterization technique

Cr2Ge2Te6 [17, 45, 50]

TC = 61k (Bulk)

TC = 30K (Bilayer, under 0.075T)

Solid State Reaction /Exfoliation

FM

Intrinsic

Insulator

Superconducting Quantum Interference Device (SQUID), Magneto-optical Kerr effect (MOKE)

CrSe2 [51]

TC = 65K (Monolayer)

CVD

FM

Intrinsic

Semiconductor

Physical Property Measurement System (PPMS, Quantum design), Raman spectroscopy, Refractive Magnetic Circular Dichroism microscopy (RMCD)

CrTe2 [52, 53]

TC = 310K (Bulk)

TC = 300K (Few layers)

TC = 200K (Monolayer)

CVD

MBE

FM

Intrinsic

Semimetal (1T)

Semiconductor

Physical Property Measurement System (PPMS, Quantum design), Vibrating Sample Magnetometer (VSM), Superconducting Quantum Interference Device (SQUID)

X-ray Absorption Spectroscopy and Magnetic Circular Dichroism, Angle-Resolved Photoemission Spectroscopy and Scanning Tunneling Microscopy

Fe3GeTe2 [57, 58, 60]

TC = 230K (Bulk)

TC = 300K (Flake, under voltage gate)

TC = 130K (Monolayer)

Solid State Reaction

/CVT/Exfoliation

FM

Intrinsic

Metal

SQUID magnetometer (MPMS, Quantum Design), Refractive Magnetic Circular Dichroism Microscopy (RMCD), Angle-Dependent Anomalous Hall Measurements

Fe5GeTe2 [63]

TC = 310K (Bulk)

TC = 270-300K (Flake)

CVT/Exfoliation

FM

Intrinsic

Unknow

Physical Property Measurement System (PPMS, Quantum design), Refractive Magnetic Circular Dichroism microscopy (RMCD)

CrI3 [16]

TC = 61K (Bulk)

TC = 45K (Monolayer)

CVT/Exfoliation

FM

Intrinsic

Insulator

Magneto-Optical Kerr effect (MOKE),

CrBr3 [69, 70]

TC = 37K (Bulk)

TC = 34K (Monolayer)

Exfoliation/MBE

FM

Intrinsic

Insulator

Polarization-Resolved Photoluminescence (PL) Spectroscopy

MnSe2 [73]

TC = 300K (Monolayer)

MBE

FM

Intrinsic

Metal

Magnetoresistance measurement, Superconducting Quantum Interference Device (SQUID)

VSe2 [76-82]

TC = 300K (Flake)

CVT/CVD/EBM

/Exfoliation

FM

Intrinsic

Metal

Superconducting Quantum Interference Device (SQUID)

MnSn [87]

TC = 54K (Monolayer)

MBE

FM

Intrinsic

Semiconductor

Superconducting Quantum Interference Device (SQUID)

FeSe2 [88, 89]

TC = 300K (Bulk)

Hydrothermal Co-Reduction Route

/CVD

FM

Intrinsic

Semiconductor

Vibrating

Sample Magnetometer (VSM)

FeTe2 [88]

TC = 300K (Bulk)

Hydrothermal Co-Reduction Route

FM

Intrinsic

Semiconductor

Vibrating

Sample Magnetometer (VSM)

FeTe [90, 91]

TC = 235K (Multilayer Hexagonal FeTe)

TN = 45K (Multilayer Tetragonal FeTe)

CVD

FM (Hexagonal)

AFM (tetragonal)

Intrinsic

Semiconductor

Vibrating

Sample Magnetometer (VSM) Superconducting Quantum Interference Device (SQUID)

MnBi8Te13 [92]

TC = 10.5K (Bulk)

Self-flux

FM

Intrinsic

Insulator

Physical Property Measurement System (PPMS, Quantum design), Magnetic Properties

Measurement System (MPMS, Quantum Design), Single-crystal neutron diffraction

α/β-In2Se3 [93-101]

Unknow

Solution method

/Exfoliation/PVD/PVT

FM

Extrinsic

Semiconductor

Theoretical calculation

MoS2 [102-115]

TC = 300K (Few-layer, fluorine adsorption)

TC = 300K (Bulk, under electron irradiation)

CVD/Exfoliation

FM

Extrinsic

Semiconductor

Theoretical calculation/ Superconducting Quantum Interference Device (SQUID)

PtSe2 [116-118]

TC ≈ 5K (~9nm thick, Defect induced)

TAC process

/Exfoliation

FM

Extrinsic

Semiconductor (Monolayer)

Semimetal (Bulk)

Theoretical calculation/ Hysteretic Magneto-Transport Response

PdSe2 [119-121]

TC = 800K (Monolayer, under uniaxial stress)

Solid State Reaction /Exfoliation

FM

Extrinsic

Semimetal

Theoretical calculation

VTe2 [125-127]

Unknow

Solid State Reaction /Liquid Exfoliation/MBE

FM

Extrinsic

Semimetal

Theoretical calculation

WS2 [128, 129, 133, 134]

TC = 470K (Monolayer)

CVD

FM

Extrinsic

Semiconductor

Theoretical calculation

SnS2 [135, 136]

TC = 31K (Monolayer, Fe-doped)

CVD/CVT/Exfoliation

FM

Extrinsic

Semiconductor

Vibrating Sample Magnetometer (VSM)

GaSe [137-142]

Unknow

vdW Epitaxy/CVT

FM

(Hole doped)

AFM

(Fe doped)

Extrinsic

Semiconductor

Theoretical calculation

GeSe [143-145]

Unknow

Solid State Reaction Bridgman method

AFM

Intrinsic

Semiconductor

Theoretical calculation

Four-Probe method

FeGa2S4 [149, 153]

TN = 160K (Bulk)

CVT

AFM

Intrinsic

Insulator

Superconducting Quantum Interference Device (SQUID)

NiGa2S4 [152, 153]

TN = 80K (Bulk)

CVT

AFM

Intrinsic

Insulator

Superconducting Quantum Interference Device (SQUID)

Fe2Ga2S5 [153]

TN = 130K (Bulk)

CVT

AFM

Intrinsic

Insulator

Superconducting Quantum Interference Device (SQUID)

FePS3 [154, 159]

TN =123K (Bulk)

TN = 118K (Monolayer)

CVT/Exfoliation

AFM

Intrinsic

Semiconductor

Raman spectroscopy

NiPS3 [155, 159]

TN = 155K (Bulk)

CVT

AFM

Intrinsic

Semiconductor

Raman spectroscopy /Superconducting Quantum Interference Device (SQUID)

MnPS3 [156, 159]

TN = 78K (Bulk)

CVT

AFM

Intrinsic

Semiconductor

Vibrating Sample Magnetometer (VSM)

MnBi2Te4 [165, 166]

TN = 25K (Bulk)

Self-fulx

AFM

Intrinsic

Insulator

Superconducting Quantum Interference Device (SQUID)

MnBi4Te7 [169]

TN = 13K (Bulk)

Self-fulx

AFM

Intrinsic

Insulator

Magnetic Property Measurement System (MPMS, Quantum Design), Neutron diffraction

MnBi6Te10 [169]

TN = 11K (Bulk)

Self-fulx

AFM

Intrinsic

Insulator

Magnetic Property Measurement System (MPMS, Quantum Design), Neutron diffraction

  • Kindly add the following references: https://arxiv.org/ftp/arxiv/papers/1608/1608.03059.pdf; Duong et al. ACS Nano 2017, 11, 11803−11830

Reply: We thank the reviewers for your gentle advice. This reference was cited in the main text.

“Various kinds of 2D materials, such as transition metal dichalcogenides (TMDs), ternary transition metal compounds (ABX3), transition metal carbide/nitride (MXene) and their potential related applications, were reported in succession, especially 2D ferromagnetism possess rapidly developing [4].”

  1. Duong, D.L.; Yun, S.J.; Lee, Y.H. van der Waals layered materials: opportunities and challenges. ACS nano 2017, 11, 11803-11830.

Reviewer 2 Report

Authors have summarized a detailed overview on the 2D van der Waals magnetic materials, which is of utmost importance for the development of future spintronic devices. The review is very interesting and very well written. However, the reviewer has few comments to be addressed.

  1. There exists a recently published detailed review on the same topic - Mongur Hossain, Biao Qin, Bo Li, Xidong Duan; Synthesis, characterization, properties and applications of two-dimensional magnetic materials, Nano Today, Volume 42, 2022, 101338, ISSN 1748-0132, https://doi.org/10.1016/j.nantod.2021.101338.
  2. There is no particular details about the synthesis methods and characterization to understand the magnetic behaviour.
  3. There could be detailed section on the ‘Magnetic Skyrmions’, which is less explained in literature.
  4. On the other hand, there has been reports on spin behaviour of large area topological insulators exhibiting spin to charge (S2C) with the inverse Edelstein effect (IEE), a conversion efficiency of λIEE  ≈  0.27 nm, in Fe/Au/Sb2Te3, originating from the topologically protected surface states (TSS) of Sb2Te3. [Spin-Charge Conversion in Fe/Au/Sb2Te3 Heterostructures as Probed By Spin Pumping Ferromagnetic Resonance, Advanced Materials Interfaces, 2021, 2101244]. There should be a section based large area growth of materials with experimental techniques, as this could be beneficial for the realization of practical applications. This could also make some difference to what has been stated in previous comment 1.
  5. Remove ‘2’ from line 8.
  6. Reference should be added in line 30.
  7. Maintain the consistency throughout the manuscript, two dimensional, 2D, ferromagnetism, FM, neel, Neel, TC, Ising ising…….
  8. Line 46, it should be orientation in low symmetry, otherwise the sentence is confusing.
  9. Line 51, …other order types. Mention type of order, otherwise the sentence is incomplete.
  10. Line 135, has to be rewritten. .. According to his are….
  11. Line 142, ‘itinerant’ should be ‘itinerant (delocalised)’, as of line 153 and vice versa.
  12. Section 2.3, there should be a general sentence defining the Neel temperature, as of similar to Curie temperature, to complete the section.
  13. Line 240, please recheck the references 35 and 36, as they are on Fe3GeTe3 and (In, Mn)As.
  14. Line 285-291, it’s a computational study, recheck and specify.
  15. The reported results are a mix of crystal and layers, even exfoliated, reviewer suggest to change the title and sub sections to ‘materials’ instead of ‘crystals’.
  16. Line 573, space between ‘nanoribbon and (diameter..).

Author Response

Reviewer #2: Authors have summarized a detailed overview on the 2D van der Waals magnetic materials, which is of utmost importance for the development of future spintronic devices. The review is very interesting and very well written. However, the reviewer has few comments to be addressed.

  • There exists a recently published detailed review on the same topic - Mongur Hossain, Biao Qin, Bo Li, Xidong Duan; Synthesis, characterization, properties and applications of two-dimensional magnetic materials, Nano Today, Volume 42, 2022, 101338, ISSN 1748-0132, https://doi.org/10.1016/j.nantod.2021.101338.

Reply: We thank the reviewers for your gentle reminder. Compared with the review article the reviewers mentioned, our article has three main differences: a) We systematically summarized the origin and theories of magnetism in detail, including exchange interaction (Ruderman–Kittel–Kasuya–Yosida (RKKY) exchange, double exchange and superexchange), magnetic domain and magnetic anisotropy, etc. b) We comprehensively classified the magnetic orderings type (ferromagnetic or antiferromagnetic) of the 2D vdW magnetic materials described. Moreover, the intrinsic/extrinsic magnetism of 2D vdW ferromagnetic/antiferromagnetic materials was reviewed in detail in main text. c) Our review focuses more on the magnetic properties of 2D vdW ferromagnetic/antiferromagnetic materials, instead of those 2D materials in a broad sense.

  • There are no particular details about the synthesis methods and characterization to understand the magnetic behaviour.

Reply: We thank the reviewers for your gentle reminder. We added the synthesis methods and characterization methods of 2D vdW magnetic materials mentioned in main text, and briefly summarized their magnetic and electronic properties with respect to their thickness in Table 1.

“We systematically summarized the recent study of 2D vdW magnetic materials respect to their thickness in Table1.”

Table 1

Transition temperature, synthesis technique, intrinsic or extrinsic magnetic orderings, electrical properties of 2D van der Waals magnetic materials.

Materials

Transition temperature

Synthesis technique

Magnetic orderings

Intrinsic/Extrinsic

Electrical properties

Characterization technique

Cr2Ge2Te6 [17, 45, 50]

TC = 61k (Bulk)

TC = 30K (Bilayer, under 0.075T)

Solid State Reaction /Exfoliation

FM

Intrinsic

Insulator

Superconducting Quantum Interference Device (SQUID), Magneto-optical Kerr effect (MOKE)

CrSe2 [51]

TC = 65K (Monolayer)

CVD

FM

Intrinsic

Semiconductor

Physical Property Measurement System (PPMS, Quantum design), Raman spectroscopy, Refractive Magnetic Circular Dichroism microscopy (RMCD)

CrTe2 [52, 53]

TC = 310K (Bulk)

TC = 300K (Few layers)

TC = 200K (Monolayer)

CVD

MBE

FM

Intrinsic

Semimetal (1T)

Semiconductor

Physical Property Measurement System (PPMS, Quantum design), Vibrating Sample Magnetometer (VSM), Superconducting Quantum Interference Device (SQUID)

X-ray Absorption Spectroscopy and Magnetic Circular Dichroism, Angle-Resolved Photoemission Spectroscopy and Scanning Tunneling Microscopy

Fe3GeTe2 [57, 58, 60]

TC = 230K (Bulk)

TC = 300K (Flake, under voltage gate)

TC = 130K (Monolayer)

Solid State Reaction

/CVT/Exfoliation

FM

Intrinsic

Metal

SQUID magnetometer (MPMS, Quantum Design), Refractive Magnetic Circular Dichroism Microscopy (RMCD), Angle-Dependent Anomalous Hall Measurements

Fe5GeTe2 [63]

TC = 310K (Bulk)

TC = 270-300K (Flake)

CVT/Exfoliation

FM

Intrinsic

Unknow

Physical Property Measurement System (PPMS, Quantum design), Refractive Magnetic Circular Dichroism microscopy (RMCD)

CrI3 [16]

TC = 61K (Bulk)

TC = 45K (Monolayer)

CVT/Exfoliation

FM

Intrinsic

Insulator

Magneto-Optical Kerr effect (MOKE),

CrBr3 [69, 70]

TC = 37K (Bulk)

TC = 34K (Monolayer)

Exfoliation/MBE

FM

Intrinsic

Insulator

Polarization-Resolved Photoluminescence (PL) Spectroscopy

MnSe2 [73]

TC = 300K (Monolayer)

MBE

FM

Intrinsic

Metal

Magnetoresistance measurement, Superconducting Quantum Interference Device (SQUID)

VSe2 [76-82]

TC = 300K (Flake)

CVT/CVD/EBM

/Exfoliation

FM

Intrinsic

Metal

Superconducting Quantum Interference Device (SQUID)

MnSn [87]

TC = 54K (Monolayer)

MBE

FM

Intrinsic

Semiconductor

Superconducting Quantum Interference Device (SQUID)

FeSe2 [88, 89]

TC = 300K (Bulk)

Hydrothermal Co-Reduction Route

/CVD

FM

Intrinsic

Semiconductor

Vibrating

Sample Magnetometer (VSM)

FeTe2 [88]

TC = 300K (Bulk)

Hydrothermal Co-Reduction Route

FM

Intrinsic

Semiconductor

Vibrating

Sample Magnetometer (VSM)

FeTe [90, 91]

TC = 235K (Multilayer Hexagonal FeTe)

TN = 45K (Multilayer Tetragonal FeTe)

CVD

FM (Hexagonal)

AFM (tetragonal)

Intrinsic

Semiconductor

Vibrating

Sample Magnetometer (VSM) Superconducting Quantum Interference Device (SQUID)

MnBi8Te13 [92]

TC = 10.5K (Bulk)

Self-flux

FM

Intrinsic

Insulator

Physical Property Measurement System (PPMS, Quantum design), Magnetic Properties

Measurement System (MPMS, Quantum Design), Single-crystal neutron diffraction

α/β-In2Se3 [93-101]

Unknow

Solution method

/Exfoliation/PVD/PVT

FM

Extrinsic

Semiconductor

Theoretical calculation

MoS2 [102-115]

TC = 300K (Few-layer, fluorine adsorption)

TC = 300K (Bulk, under electron irradiation)

CVD/Exfoliation

FM

Extrinsic

Semiconductor

Theoretical calculation/ Superconducting Quantum Interference Device (SQUID)

PtSe2 [116-118]

TC ≈ 5K (~9nm thick, Defect induced)

TAC process

/Exfoliation

FM

Extrinsic

Semiconductor (Monolayer)

Semimetal (Bulk)

Theoretical calculation/ Hysteretic Magneto-Transport Response

PdSe2 [119-121]

TC = 800K (Monolayer, under uniaxial stress)

Solid State Reaction /Exfoliation

FM

Extrinsic

Semimetal

Theoretical calculation

VTe2 [125-127]

Unknow

Solid State Reaction /Liquid Exfoliation/MBE

FM

Extrinsic

Semimetal

Theoretical calculation

WS2 [128, 129, 133, 134]

TC = 470K (Monolayer)

CVD

FM

Extrinsic

Semiconductor

Theoretical calculation

SnS2 [135, 136]

TC = 31K (Monolayer, Fe-doped)

CVD/CVT/Exfoliation

FM

Extrinsic

Semiconductor

Vibrating Sample Magnetometer (VSM)

GaSe [137-142]

Unknow

vdW Epitaxy/CVT

FM

(Hole doped)

AFM

(Fe doped)

Extrinsic

Semiconductor

Theoretical calculation

GeSe [143-145]

Unknow

Solid State Reaction Bridgman method

AFM

Intrinsic

Semiconductor

Theoretical calculation

Four-Probe method

FeGa2S4 [149, 153]

TN = 160K (Bulk)

CVT

AFM

Intrinsic

Insulator

Superconducting Quantum Interference Device (SQUID)

NiGa2S4 [152, 153]

TN = 80K (Bulk)

CVT

AFM

Intrinsic

Insulator

Superconducting Quantum Interference Device (SQUID)

Fe2Ga2S5 [153]

TN = 130K (Bulk)

CVT

AFM

Intrinsic

Insulator

Superconducting Quantum Interference Device (SQUID)

FePS3 [154, 159]

TN =123K (Bulk)

TN = 118K (Monolayer)

CVT/Exfoliation

AFM

Intrinsic

Semiconductor

Raman spectroscopy

NiPS3 [155, 159]

TN = 155K (Bulk)

CVT

AFM

Intrinsic

Semiconductor

Raman spectroscopy /Superconducting Quantum Interference Device (SQUID)

MnPS3 [156, 159]

TN = 78K (Bulk)

CVT

AFM

Intrinsic

Semiconductor

Vibrating Sample Magnetometer (VSM)

MnBi2Te4 [165, 166]

TN = 25K (Bulk)

Self-fulx

AFM

Intrinsic

Insulator

Superconducting Quantum Interference Device (SQUID)

MnBi4Te7 [169]

TN = 13K (Bulk)

Self-fulx

AFM

Intrinsic

Insulator

Magnetic Property Measurement System (MPMS, Quantum Design), Neutron diffraction

MnBi6Te10 [169]

TN = 11K (Bulk)

Self-fulx

AFM

Intrinsic

Insulator

Magnetic Property Measurement System (MPMS, Quantum Design), Neutron diffraction

  • There could be detailed section on the ‘Magnetic Skyrmions’, which is less explained in literature.

Reply: We are grateful to this constructive suggestion. We added a section 5.4 Magnetic Skyrmions” in main text, in which magnetic Skyrmions were explained in detailed.

5.4 Magnetic Skyrmions

        Magnetic skymions, sometimes also called vortices [186] or vortex-like [187] configurations, was firstly predicted theoretically [186, 188, 189] and characterized experimentally by means of neutron scattering in momentum space and microscopy techniques in real space in recent years [190-192]. Magnetic skymions are nanometre-sized quasiparticles [193], which defined as the local whirl of the spin configuration in a magnetic material. Compare with single-domain, magnetic skyrmions possessed more energetical stability (per unit volum) and allow the existence of discrete magnetic states. Therefore, magnetic skyrmions, acted as bits to store information, are promising to be used in future high-performance memory [194-196] and logic [197, 198] devices. Besides, dynamical magnetic skyrmions exhibit impressive robust breathing feature, which means meaking the skyrmion-based microwave applications a reality is achievable [199]. However, there are some intriguing and important questions needed to be investigated competitively, such as the creation and annihilation processes, the control of size of magnetic skyrmions, more accurate understanding of the role of defects in magnetic skyrmions, etc.”

  1. Bogdanov, A.; Rößler, U. Chiral symmetry breaking in magnetic thin films and multilayers. Physical review letters 2001, 87, 037203.
  2. Iwasaki, J.; Mochizuki, M.; Nagaosa, N. Current-induced skyrmion dynamics in constricted geometries. Nat. Nanotechnol. 2013, 8, 742-747.
  3. Roessler, U.K.; Bogdanov, A.; Pfleiderer, C. Spontaneous skyrmion ground states in magnetic metals. Nature 2006, 442, 797-801.
  4. Dupé, B.; Hoffmann, M.; Paillard, C.; et al. Tailoring magnetic skyrmions in ultra-thin transition metal films. Nat. Commun. 2014, 5, 1-6.
  5. Romming, N.; Hanneken, C.; Menzel, M.; et al. Writing and deleting single magnetic skyrmions. Science 2013, 341, 636-639.
  6. Mühlbauer, S.; Binz, B.; Jonietz, F.; et al. Skyrmion lattice in a chiral magnet. Science 2009, 323, 915-919.
  7. Hsu, P.-J.; Kubetzka, A.; Finco, A.; et al. Electric-field-driven switching of individual magnetic skyrmions. Nat. Nanotechnol. 2017, 12, 123-126.
  8. Sondhi, S.L.; Karlhede, A.; Kivelson, S.; et al. Skyrmions and the crossover from the integer to fractional quantum Hall effect at small Zeeman energies. Phys. Rev. B: Condens. Matter 1993, 47, 16419.
  9. Sampaio, J.; Cros, V.; Rohart, S.; et al. Nucleation, stability and current-induced motion of isolated magnetic skyrmions in nanostructures. Nat. Nanotechnol. 2013, 8, 839-844.
  10. Tomasello, R.; Martinez, E.; Zivieri, R.; et al. A strategy for the design of skyrmion racetrack memories. Sci. Rep. 2014, 4, 1-7.
  11. Kang, W.; Huang, Y.; Zheng, C.; et al. Voltage controlled magnetic skyrmion motion for racetrack memory. Sci. Rep. 2016, 6, 1-11.
  12. Zhou, Y.; Ezawa, M. A reversible conversion between a skyrmion and a domain-wall pair in a junction geometry. Nat. Commun. 2014, 5, 1-8.
  13. Zhang, X.; Ezawa, M.; Zhou, Y. Magnetic skyrmion logic gates: conversion, duplication and merging of skyrmions. Sci. Rep. 2015, 5, 1-8.
  14. Zhou, Y.; Iacocca, E.; Awad, A.A.; et al. Dynamically stabilized magnetic skyrmions. Nat. Commun. 2015, 6, 1-10.

  • On the other hand, there has been reports on spin behaviour of large area topological insulators exhibiting spin to charge (S2C) with the inverse Edelstein effect (IEE), a conversion efficiency of λIEE ≈ 0.27 nm, in Fe/Au/Sb2Te3, originating from the topologically protected surface states (TSS) of Sb2Te3. [Spin-Charge Conversion in Fe/Au/Sb2Te3 Heterostructures as Probed By Spin Pumping Ferromagnetic Resonance, Advanced Materials Interfaces, 2021, 2101244]. There should be a section based large area growth of materials with experimental techniques, as this could be beneficial for the realization of practical applications. This could also make some difference to what has been stated in previous comment 1.

Reply: We sincerely thank the reviewer for this constructive suggestion. We added a section “5.1 Synthesis techniques” in main text, in which we introduce the experimental techniques of large area growth of 2D vdW materials.

5.1 Synthesis

Large area growth of is one of the main challenges of 2D materials, which impedes the practical application of 2D materials. Until now, mechanical exfoliation is still one of the most widely adopted methods to obtained 2D materials. Through mechanical exfoliation, people obtained 2D graphene for the first time [170]. Generally, mechanical exfoliation can acquire high quality 2D materials (Figure 8a) [171]. However, this method is inefficient and unable to integrate with industry. More importantly, mechanical exfoliation is not suitable for all kinds of 2D metails. Recently, Huang et al. propose a new strategy, in which the mechanical exfoliation assisted by an Au adhesion layer with covalent-like quasi bonding to a layered crystal provides access to a broad spectrum of large-area monolayer materials. Significantly, such an approach is applicable a large number of 2D materials (Figure 8b) [172]. Similarly, Moon et al. developed a layer-engineered exfoliation (LEE) approach to obtained extraordinarily large-size and high-density graphene from natural graphite. In this method, researchers directly evaporated a selective metal film on to a bulk graphite precleaved on the bule tape to o exfoliate large area graphene with a selective number of layers. Such a LEE approach paves the way for the development of a manufacturing-scale process for future applications based on 2D heterostructure (Figure 8c) [173]. Liquid exfoliation is another general method to synthesis high quality 2D materials. Coleman et al. demonstrated that several layered materials such as MoS2, MoSe2, WS2, BN can be efficiently dispersed in common solvents and can be deposited as individual flakes or formed into film (Figure 8d) [174]. In this method, the choice of solvent is the key factor. Zeng et al. created an electrochemical way to fabrication few-layer-thick 2D materials such as BN, NbSe2, WSe2. In this method, the lithium intercalation of the layered material was performed in a test cell with the Li foil as anode and 1 M LiPF6 dissolved in the mixture of ethyl carbonate (EC) and dimethyl carbonate (DMC) (1:1 in volume ratio) as electrolyte. The layered bulk material was prepared as cathode by adding acetylene black and poly (vinylidene fluoride) (PVDF) binder dispersed in N-methylpyrrolidone (NMP) solutions. After the discharge process, the lithium-intercalated sample was washed with acetone to remove the residual electrolyte (LiPF6), followed by exfoliation and sonication in N2-saturated Milli-Q water in a closed vial (Figure 8e) [175]. Chemical vapor deposition (CVD) is also one of the most widely adopted methods to synthesize 2D materials, which is considered as one of the most suitable methods for large-area growth of 2D materials. In 2006, Somani et al. experimentally demonstrated that planer few-layer nano-graphenes (PFLGs) can be synthesized by simple thermal CVD method for the first time [176]. In the past several years, the growth of 2D materials based on CVD method have been extensively developed. Via CVD method Many kinds of high quality 2D crystals, continuous films, and even heterostructures have been grown with a large-scale production capability and good controllability, which are of great significance of bringing practical applications to fruition (Figure 8f) [177]. Molecular beam epitaxy (MBE) in another efficient way to synthesize ultrathin films, which exhibits great superiority in controlling the thickness of materials. In previous report, some 2D materials such as CrBr3 [70] (Figure 8g), CrTe2 [53], MnSe2 [73] and MnSn [87], were successfully synthesized via molecular beam epitaxy (MBE). However, the costly facilities and complex processes hinder its development in the growth of 2D materials.”

  1. Zhang, X.; Lu, Q.; Liu, W.; et al. Room-temperature intrinsic ferromagnetism in epitaxial CrTe2 ultrathin films. Nat. Commun. 2021, 12, 1-9.
  2. Chen, W.; Sun, Z.; Wang, Z.; et al. Direct observation of van der Waals stacking–dependent interlayer magnetism. Science 2019, 366, 983-987.
  3. O'Hara, D.J.; Zhu, T.; Trout, A.H.; et al. Room Temperature Intrinsic Ferromagnetism in Epitaxial Manganese Selenide Films in the Monolayer Limit. Nano Lett. 2018, 18, 3125-3131.
  4. Yuan, Q.-Q.; Guo, Z.; Shi, Z.-Q.; et al. Ferromagnetic MnSn monolayer epitaxially grown on silicon substrate. Chinese Phys. Lett. 2020, 37, 077502.
  5. Novoselov, K.S.; Geim, A.K.; Morozov, S.V.; et al. Electric Field Effect in Atomically Thin Carbon Films. Science 2004, 306, 666.
  6. Yi, M.; Shen, Z. A review on mechanical exfoliation for the scalable production of graphene. J. Mater. Chem. A 2015, 3, 11700-11715.
  7. Huang, Y.; Pan, Y.-H.; Yang, R.; et al. Universal mechanical exfoliation of large-area 2D crystals. Nat. Commun. 2020, 11, 2453.
  8. Moon, J.-Y.; Kim, M.; Kim, S.-I.; et al. Layer-engineered large-area exfoliation of graphene. Sci. Adv. 2020, 6, eabc6601.
  9. Coleman, J.N.; Lotya, M.; O’Neill, A.; et al. Two-dimensional nanosheets produced by liquid exfoliation of layered materials. Science 2011, 331, 568-571.
  10. Zeng, Z.; Sun, T.; Zhu, J.; et al. An effective method for the fabrication of few‐layer‐thick inorganic nanosheets. Angew. Chem. Int. Ed. 2012, 51, 9052-9056.
  11. Somani, P.R.; Somani, S.P.; Umeno, M. Planer nano-graphenes from camphor by CVD. Chem. Phys. Lett. 2006, 430, 56-59.
  12. Cai, Z.; Liu, B.; Zou, X.; et al. Chemical vapor deposition growth and applications of two-dimensional materials and their heterostructures. Chem. Rev. 2018, 118, 6091-6133.

Figure 8. a) An illustrative procedure of micromechanical cleavage of graphene based on transparent tape. Reused with permission from [171] b) Schematic of the exfoliation process assisted by an Au adhesion. Reused with permission from [172] c) Schematic illustration of layer-engineered exfoliation (LEE) large-area graphene exfoliation technique. Reused with permission from [173] d) Photographs of dispersions of MoS2 (in N-methyl-pyrrolidone (NMP)), WS2 (in N-methyl-pyrrolidone (NMP)), and BN (in isopropanol (IPA)). Reused with permission from [174] e) The electrochemical lithium intercalation process to produce 2D nanosheets from the layered bulk material. Reused with permission from [175] f) Overview of the CVD growth of 2D materials, from single crystals to continuous films, and further to 2D material based heterostructures, as well as their device applications. Reused with permission from [177] g) Illustration of MBE set-up for the growth of CrTe2 films on graphene. Reused with permission from [53]

  • Remove ‘2’ from line 8.

Reply: Thank you for your careful reading of our manuscript. We apologize for any confusion caused and appreciate the valuable suggestions. We have corrected it in main text.

  • Reference should be added in line 30.

Reply: We thank the reviewers for your gentle reminder. We added the references in the sentence that you mentioned.

“Although the research on low dimensional ferromagnets can be traced back to early 1970s, due to the limitation of materials synthesis technology, the investigation of ferromagnets has mostly focused on bulk materials for quite a long time [1-3].”

  1. Cullity, B.D.; Graham, C.D. Introduction to magnetic materials; John Wiley & Sons: Canada, 2011; pp. 10-15
  2. Kittel, C. Introduction to Solid State Physics; Wiley: New York, 1990; pp. 90-93
  3. Jonker, G.; Van Santen, J. Ferromagnetic compounds of manganese with perovskite structure. physica 1950, 16, 337-349.

  • Maintain the consistency throughout the manuscript, two dimensional, 2D, ferromagnetism, FM, neel, Neel, TC, Ising ising……

Reply: We are grateful to this constructive suggestion and sorry for our mistakes. We have corrected it in main text, which makes the whole article more holistic.

  • Line 46, it should be orientation in low symmetry, otherwise the sentence is confusing.

Reply: We thank the reviewers for your kind advice. We want to express that in low dimensional systems, the low symmetry of 2D materials is necessary to realize their ferromagnetism. We rewrote this sentence in the following style.

“It indicates that the low symmetry of 2D materials is necessary to realize ferromagnetism in low dimensionality system.”

  • Line 51, …other order types. Mention type of order, otherwise the sentence is incomplete.

Reply: We are grateful to this constructive suggestion, and we rewrote this sentence to make it more complete.

“Meanwhile, the existence of internal enhancement and new potential fluctuations, theoretically, may also result in other order types, such as superfluid order, superconducting order and topologic order [11, 15].”

  1. Mermin, N.D.; Wagner, H. Absence of Ferromagnetism or Antiferromagnetism in One- or Two-Dimensional Isotropic Heisenberg Models. Phys. Rev. Lett. 1966, 17, 1133-1136.
  2. Nelson, D.; Piran, T.; Weinberg, S. Statistical mechanics of membranes and surfaces; World Scientific: Singapore, 2004; pp. 19-45
  3. Nelson, D.; Peliti, L. Fluctuations in membranes with crystalline and hexatic order. Journal de physique 1987, 48, 1085-1092.
  4. Le Doussal, P.; Radzihovsky, L. Self-consistent theory of polymerized membranes. Physical review letters 1992, 69, 1209.
  5. Kosterlitz, J.M.; Thouless, D.J. Ordering, metastability and phase transitions in two-dimensional systems. J. Phys. C: Solid State Phys. 1973, 6, 1181.

  • Line 135, has to be rewritten. .. According to his are….

Reply: Thank you for your careful reading of our manuscript. We apologize for any confusion caused and appreciate the valuable reminder. This is entirely caused by our carelessness, and we rewrote this sentence as below.

“P.-G. de Gennes systematically explored the effects of double exchange in magnetic crystals, according to his conclusion, the special form of the double exchange coupling is such that all antiferromagnetic (and also all ferrimagnetic) spin arrangements are distorted as soon as some Zener carriers are present [30].”

  • Line 142, ‘itinerant’ should be ‘itinerant (delocalised)’, as of line 153 and vice versa.

Reply: We thank the reviewers for your gentle advice. We corrected this mistake in main text.

“They find that there exist in CrO2 both localized and itinerant (delocalized) d electrons, resulting in ferromagnetic ordering due to double exchange similar to colossal magnetoresistance manganates (Figure 1b) [31].”

  1. Korotin, M.; Anisimov, V.; Khomskii, D.; et al. CrO2: a self-doped double exchange ferromagnet. Phys. Rev. Lett. 1998, 80, 4305.

“The difference between superexchange and double exchange is the occupancy of the d-shell of the two metal ions is the same or differs by two, while the electrons are itinerant (delocalized) in double exchange.”

  • Section 2.3, there should be a general sentence defining the Neel temperature, as of similar to Curie temperature, to complete the section.

Reply: We are grateful to this constructive suggestion, and we added a general sentence as below to define the Neel temperature too.

“Similarly, Néel temperature is the highest temperature that antiferromagnetic phase exist stably, otherwise, antiferromagnetic phase will transform to paramagnetic phase when the temperature higher than Néel temperature.”

  • Line 240, please recheck the references 35 and 36, as they are on Fe3GeTe2 and (In, Mn)As.

Reply: Thank you for your careful reading of our manuscript. We apologize for any confusion caused and appreciate the valuable suggestions. These two references should be added after the previous one sentence, and we corrected it. Because Cr2Ge2Te6 is a insulator, so we want to express that detection method of Fe3GeTe3 and (In, Mn) As is not suitable for Cr2Ge2Te6. And later in section “3.1.2 Fe3GeTe2, Fe5GeTe2”, we cited the references of Fe3GeTe2 again.

“Moreover, Tan et al. experimentally investigated the anomalous Hall effect on single crystalline FGT nanoflakes and show that their magnetic properties are highly dependent on thickness. Importantly, a hard magnetic phase with large coercivity and near square-shaped hysteresis loop will occur when the thickness is reduced to less than 200 nm. These characteristics are accompanied by strong perpendicular magnetic anisotropy, making vdW FGT a ferromagnetic metal suitable for vdW heterostructure-based spintronics [48].”

  1. Tan, C.; Lee, J.; Jung, S.-G.; et al. Hard magnetic properties in nanoflake van der Waals Fe3GeTe2. Nat. Commun. 2018, 9, 1554.

  • Line 285-291, it’s a computational study, recheck and specify.

Reply: We greatly appreciate the reviewer with pointing out this mistake, and we included the details of the type of this study.

  • The reported results are a mix of crystal and layers, even exfoliated, reviewer suggest to change the title and sub sections to ‘materials’ instead of ‘crystals’.

Reply: We are grateful to this constructive suggestion, and we changed the title and sub sections from ‘crystals’ to ‘materials.

  • Line 573, space between ‘nanoribbon and (diameter..).

Reply: We greatly appreciate the reviewer with pointing out this mistake, which was corrected in main text.

Reviewer 3 Report

I suggest some corrections.

General:

-What about magnetic hysteresis for  the materials described, authors should add a comment?

-Does temperature affect the magnetic properties of the materials described, comment?

- Introduction; line 37: Add text: “These materials are particularly important in highly sensitive magnetic measurement methods:

-Matko, V., MILANOVIČ, M. High resolution switching mode inductance-to-frequency converter with temperature compensation. Sensors, ISSN 1424-8220, 2014, vol. 14, no. 10, str. 19242-19259. https://www.mdpi.com/1424-8220/14/10/19242

-Yang, S.; Tan, M.; Yu, T.; Li, X.; Wang, X.; Zhang, J. Hybrid Reduced Graphene Oxide with Special Magnetoresistance for Wireless Magnetic Field Sensor. Nano-Micro Letters 2020, 12, 1-14, doi:10.1007/s40820-020-0403-9.

-Matko V., Šafarič R. Major improvements of quartz crystal pulling sensitivity and linearity using series reactance.  Sensors, 2009, 9, 10, 8263-8270.  https://www.ncbi.nlm.nih.gov/pmc/articles/PMC3292106/

    Athors should include the references above in the paper.

Author Response

Reviewer #3: I suggest some corrections.

General:

  • What about magnetic hysteresis for the materials described, authors should add a comment?

Reply: Thanks a lot for your question. For a part of 2D vdW magnetic materials described, they exhibit clear hysteresis loop at low temperature even above room temperature. For other 2D vdW magnetic materials described, owing to the weak magnetism, hysteresis loop can not be observed significantly, so the researchers investigate their magnetic properties via special measurement method, such as raman spectroscopy, polarization-resolved photoluminescence (PL) spectroscopy, reflective magnetic circular dichroism (RMCD) and the magneto-optical Kerr effect (MOKE) microscopy.

  • Does temperature affect the magnetic properties of the materials described, comment?

Reply: Thanks a lot for your question. Absolutely, temperature will greatly affect the magnetic properties of materials described. Curie temperature (TC) and Néel temperature (TN), which were mentioned in section 2.3, is two of the most important parameters of magnetic materials. Generally speaking, with the decrease of temperature, materials exhibit stronger magnetism. One of the main challenges of 2D vdW magnetic materials is the weak magnetism and low TC (TN), hindering the practical application of 2D vdW magnetic materials, which still need take more efforts to investigate on.

(3) Introduction; line 37: Add text: “These materials are particularly important in highly sensitive magnetic measurement methods:

-Matko, V., MILANOVIČ, M. High resolution switching mode inductance-to-frequency converter with temperature compensation. Sensors, ISSN 1424-8220, 2014, vol. 14, no. 10, str. 19242-19259. https://www.mdpi.com/1424-8220/14/10/19242

-Yang, S.; Tan, M.; Yu, T.; Li, X.; Wang, X.; Zhang, J. Hybrid Reduced Graphene Oxide with Special Magnetoresistance for Wireless Magnetic Field Sensor. Nano-Micro Letters 2020, 12, 1-14, doi:10.1007/s40820-020-0403-9.

-Matko V., Šafarič R. Major improvements of quartz crystal pulling sensitivity and linearity using series reactance. Sensors, 2009, 9, 10, 8263-8270. https://www.ncbi.nlm.nih.gov/pmc/articles/PMC3292106/

Athors should include the references above in the paper.

Reply: We thank the reviewers for your gentle advice, and all references were cited in the main text.

“These materials are particularly important in highly sensitive magnetic measurement methods [5-7].”

  1. Matko, V.; Milanović, M. High resolution switching mode inductance-to-frequency converter with temperature compensation. Sensors 2014, 14, 19242-19259.
  2. Yang, S.; Tan, M.; Yu, T.; et al. Hybrid reduced graphene oxide with special magnetoresistance for wireless magnetic field sensor. Nanomicro Lett. 2020, 12, 1-14.
  3. Matko, V.; Šafarič, R. Major improvements of quartz crystal pulling sensitivity and linearity using series reactance. Sensors 2009, 9, 8263-8270.

Reviewer 4 Report

In this review article, He et al. discussed the current advances in intrinsic and induced ferromagnetism and/or antiferromagnetism, physical properties, device fabrication and potential applications of 2D van der Waals Magnetic Crystals. I have a few concerns about this work and are listed below.

  1. The authors can include the proper citation of Mervin -Wagner theorem and the origin of anharmonicity and ripple formation in 2D materials, which actually gives stability to 2D materials (Nature Materials, 2007, 6, 858-861).
  2. The name of Charles Kittel is wrongly written in some places (mainly on Page 3). The authors can think of cross-checking and correcting this. 
  3. The representation of van der Waals crystals are not proper (E.g. In the case of Cr2Ge2Te6, the number should be in the subscript position). 
  4. The authors should include the schematics of all the structures (top and side views) discussed in the manuscript. I could see the bulk structure rather than 2D structures in Fig 2 and also the CrI3 structure in Fig. 3. 
  5. It would be if the authors can include the details of what type of studies has been carried out (experimental/theoretical) for the property manipulation of 2D vdw heterostructures. 

Author Response

Reviewer #4: In this review article, He et al. discussed the current advances in intrinsic and induced ferromagnetism and/or antiferromagnetism, physical properties, device fabrication and potential applications of 2D van der Waals Magnetic Crystals. I have a few concerns about this work and are listed below.

  • The authors can include the proper citation of Mervin -Wagner theorem and the origin of anharmonicity and ripple formation in 2D materials, which actually gives stability to 2D materials (Nature Materials, 2007, 6, 858-861).

Reply: We thank the reviewers for your gentle suggestion. We included the citation of Mervin -Wagner theorem and the origin of anharmonicity and ripple formation in 2D materials in the main text.

“According to Mermin-Wagner theorem, the short-range isotropic interaction cannot realize long-range continuous rotational symmetry in low-dimensional systems, as demonstrated by 2D isotropic Heisenberg model. This is mainly owing to the enhancement of 2D fluctuations, induced the unsustainable order and symmetry breaking. Nevertheless, some earlier theoretical researchs based on 2D membrane revealed that the anharmonic coupling can suppress such fluctuations [12-14].”

  1. Nelson, D.; Piran, T.; Weinberg, S. Statistical mechanics of membranes and surfaces; World Scientific: Singapore, 2004; pp. 19-45
  2. Nelson, D.; Peliti, L. Fluctuations in membranes with crystalline and hexatic order. Journal de physique 1987, 48, 1085-1092.
  3. Le Doussal, P.; Radzihovsky, L. Self-consistent theory of polymerized membranes. Physical review letters 1992, 69, 1209.

  • The name of Charles Kittel is wrongly written in some places (mainly on Page 3). The authors can think of cross-checking and correcting this.

Reply: Thank you for your careful reading of our manuscript. We apologize for any confusion caused and greatly appreciate the reviewer with pointing out this mistake, which was corrected in main text.

  • The representation of van der Waals crystals are not proper (E.g. In the case of Cr2Ge2Te6, the number should be in the subscript position).

Reply: We thank the reviewers for your gentle reminder. By comparing the word version with the PDF version, we find that it seems to be a problem with the PDF version. In the word version, the number of chemical formula of van der Waals crystals are all in the subscript position.

  • The authors should include the schematics of all the structures (top and side views) discussed in the manuscript. I could see the bulk structure rather than 2D structures in Fig 2 and also the CrI3 structure in Fig. 3.

Reply: We thank the reviewers for your gentle advice. We have added the schematics of all the structures (top and side views) in Fig 2, Fig 3 and so is Fig 4. For the AGa2S4 (A = Ni, Fe) and Fe2Ga2S5, they are quasi-2D triangular Heisenberg antiferromagnets, which are the 2D materials in broad sense.

Figure 2. a) Crystal structure of Cr2Ge2Te6. Reused with permission from [46, 47] b) Structure of monolayer FGT. The left panel shows the view along [001]; the right panel shows the view along. Reused with permission from [59] c) Temperature-dependent domain imaging of Fe3GeTe2. Reused with permission from [219] d) Criticality analysis for FGT flakes of different thicknesses. Reused with permission from [60] e) Crystal structure and real space STEM image and spectroscopic maps of Fe5-xGeTe2. The left panel is the average crystal structure obtained from single crystal X-ray diffraction data where Fe (1) and Ge are split sites; the right panel is a Z-contrast HAADF image with simulation overlaid in the region outlined by white. EELS chemical mapping of local structure. Reused with permission from [63] f) Magnetic characterization of bulk single crystals and polycrystalline samples. Temperature dependence of the magnetization upon cooling in an applied field H // c (Mc) and H⊥c (Mab) for single crystals as well as data for the polycrystalline sample (upper panel). Normalized lattice parameters revealing a strong magnetoelastic coupling near 100 K (lower panel). Reused with permission from [63]

Figure 3. a) Top view of monolayer CrI3, where Cr atoms (red balls) form a honeycomb structure in an edge-sharing octahedral coordination by six I atoms (blue balls), and side view of bilayer CrI3 of the rhombohedral stacking order. Reused with permission from [67] b) MCD versus magnetic field (left) at three representative doping levels at 4 K (top panel) and 50 K (bottom panel); MCD versus magnetic field at 4 K at representative gate voltages (right). Reused with permission from [68] c) The hysteresis loop for bilayer CrBr3. Reused with permission from [69] d) Crystal structure diagrams of α-MnSe (111) and monolayer 1T-MnSe2. Reused with permission from [73] e & f) Schematic illustration of the electrochemical setup for cathode exfoliation of VSe2 using tetra propylammonium cation (TPA) as the intercalant. Inset: photograph showing a small piece of VSe2 crystal clamped by two titanium plates (left), visible expansion of VSe2 cathode with a fluffy shape (right); M–H hysteresis loop of bare and passivated VSe2 flakes at 300 K. Reused with permission from [84] g) Structure determination and the stacking rule in MnBi2nTe3n+1. Reused with permission from [92]

Figure 4. a) Crystal structure of the Fe–In2Se3 monolayer. Red, yellow, and blue balls represent In, Se, and Fe atoms, respectively. Reused with permission from [96] b) The spin-resolved charge density isosurface (isosurface value at 0.002 e/Å3) of TM-doped MoS2 monolayer at 8 % impurity concentration. Reused with permission from [114] c) Crystal structure of monolayer PtSe2 (left panel, top: unit cell, bottom: top view); band dispersions along the K-G-M-K direction from first-principles calculations. The colour and line width distinguish the contribution from Pt and Se (right panel, Red for Se and green for Pt). Reused with permission from [116] d) Optical and AFM antiferromagnetic images of a completed PtSe2 device and its crystal (upper panel). Magnetic field dependence of ΔR measured from device E (shown in upper panel) at T = 1.5 K and VSD = 10 mV. The red (black) arrow represents the sweep direction from positive (negative) to negative (positive) values. Reused with permission from [7]

  • It would be if the authors can include the details of what type of studies has been carried out (experimental/theoretical) for the property manipulation of 2D vdw heterostructures.

Reply: We are grateful to this constructive suggestion and apologize for any confusion we caused. In the section 5.5 Properties manipulation of the main text, for some ambiguous description in previous versions, we included the details of the type of study to help readers have a better understanding of the content.

“Yang et al. experimentally demonstrated a novel two‐step tellurium flux strategy to introduce magnetic Cr atoms into layered semimetal Td‐WTe2 to realize highly tunable near-room temperature ferromagnetism, in which the semimetal properties of the Cr‐doped layered Td–WTe2 were still preserved intriguingly [200].”

“Recently, Ge et al. reported the magnetic moments induced by atomic vacancies in transition metal dichalcogenide PtSe2 flakes and they experimentally demonstrated the presence of nearly thickness‐dependent localized magnetic moments is induced by Pt‐vacancy defects in air‐stable PtSe2 flakes [202].”

“The emergent ferromagnetism near three-quarters filling in twisted bilayer graphene [203, 204] as well as the tunable correlated chern insulator and ferromagnetism in an ABC-trilayer graphene/hexagonal boron nitride (ABC-TLG/hBN) moiré superlattice [205] experimentally verified suggested that the layer-engineering may be an effective method to achieve the ambient temperature application of 2D ferromagnetism in the future. The experimental research of Zhang et al. indicated that proximity-coupling will significantly induce the enhancement of coercive field and Curie temperature in 2D van der Waals heterostructures [206].”

“Moreover, Li et al. also experimentally achieved strong ferromagnetism via breathing lattices in atomically thin cobaltites, which indicated that strain engineering and interface engineering are potential feasible ways to improve the properties of 2D ferromagnetic materials”

“Additionally, Zhang et al. experimentally relized gigantic current control of coercive field and magnetic memory based on nanometer-thin ferromagnetic van der Waals Fe3GeTe2 (FGT) which perhaps opens up a fascinating avenue of electrical modulation and spintronic applications using 2D magnetic vdw materials [209].”

“Recently, Wang et al. experimentally discovered the colossal magnetoresistance without mixed valence in a layered phosphide crystal EuCd2P2, in which the absence of direct Cd‐Cd bonds in the structure of EuCd2P2 seems to be beneficial to the CMR [210].”

Round 2

Reviewer 2 Report

The manuscript has now been significantly improved with the detailed revision in the revised draft. All the comments has been addressed with a proper explanation and appropriate change in the draft. The manuscript is now suitable for publication. 

Author Response

Thank you very much for your constructive suggestions